# Cerebellar and hepatic alterations in ACBD5-deficient mice are associated with unexpected, distinct alterations in cellular lipid homeostasis

Warda Darwisch[1,7], Marino von Spangenberg[1,7], Jana Lehmann[1], Öznur Singin[1], Geralt Deubert[1], Sandra Kühl[1], Johannes Roos[1], Heinz Horstmann[2], Christoph Körber[2], Simone Hoppe[2], Hongwei Zheng[2], Thomas Kuner[2], Mia L. Pras-Raves[3,4], Antoine H. C. van Kampen[4,5], Hans R. Waterham[3], Kathrin V. Schwarz[6], Jürgen G. Okun[6], Christian Schultz[1], Frédéric M. Vaz [3] & Markus Islinger [1✉]

ACBD5 deficiency is a novel peroxisome disorder with a largely uncharacterized pathology. ACBD5 was recently identified in a tethering complex mediating membrane contacts between peroxisomes and the endoplasmic reticulum (ER). An ACBD5-deficient mouse was analyzed to correlate ACBD5 tethering functions with the disease phenotype. ACBD5-deficient mice exhibit elevated very long-chain fatty acid levels and a progressive cerebellar pathology. Liver did not exhibit pathologic changes but increased peroxisome abundance and drastically reduced peroxisome-ER contacts. Lipidomics of liver and cerebellum revealed tissue-specific alterations in distinct lipid classes and subspecies. In line with the neurological pathology, unusual ultra-long chain fatty acids (C > 32) were elevated in phosphocholines from cerebelli but not liver indicating an organ-specific imbalance in fatty acid degradation and elongation pathways. By contrast, ether lipid formation was perturbed in liver towards an accumulation of alkyldiacylglycerols. The alterations in several lipid classes suggest that ACBD5, in addition to its acyl-CoA binding function, might maintain peroxisome-ER contacts in order to contribute to the regulation of anabolic and catabolic cellular lipid pathways.

[1] Neuroanatomy, Mannheim Center for Translational Neuroscience, Medical Faculty Mannheim, Heidelberg University, Ludolf-Krehl Str. 13-17, 68167 Mannheim, Germany. [2] Functional Neuroanatomy, Heidelberg University, Im Neuenheimer Feld 307, 69120 Heidelberg, Germany. [3] Laboratory of Genetic Metabolic Diseases, Amsterdam University Medical Center, Meibergdreef 9, 1105 AZ Amsterdam, The Netherlands. [4] Bioinformatics Laboratory, Department of Clinical Epidemiology, Amsterdam University Medical Center, Meibergdreef 9, Location J1B-208, 1100 DE Amsterdam, The Netherlands. [5] Biosystems Data Analysis, Swammerdam Institute for Life Sciences, University of Amsterdam, Sciencepark 904, 1098 XH Amsterdam, The Netherlands. [6] Dietmar-Hopp-Stoffwechselzentrum, Heidelberg University Hospital, Im Neuenheimer Feld 669, 69120 Heidelberg, Germany. [7]These authors contributed equally: Warda Darwisch, Marino von Spangenberg. ✉email: markus.islinger@medma.uni-heidelberg.de

Organelle membrane contact sites are considered to be important hubs for intracellular metabolite exchange and signaling network regulation[1]. Recently, we and others discovered that the tail-anchored membrane protein acyl-CoA binding domain containing protein 5 (ACBD5) is localized at peroxisomes by inserting its C-terminus in the peroxisomal membrane while its acyl-CoA binding domain faces the cytosol[2–4]. We further revealed that ACBD5 is a tethering protein binding the endoplasmic reticulum (ER) resident proteins VAPA and VAPB[5,6]. Interaction between ACBD5 and VAP induces membrane contact site formation between peroxisomes and the ER. Although such close appositions between both organelles have been observed for decades, their functional significance remains poorly understood[7]. ACBD5-deficient patients develop a severe, progressive retino- and neuropathy beginning during early infancy[8,9] suggesting a causal relationship between peroxisome-ER contact site formation and disease pathogenesis. Peroxisomes play an essential role in the metabolism of cellular lipids[10]. Correspondingly, ACBD5-deficient patients exhibit elevated levels of very long-chain fatty acids (VLCFA) in plasma and reduced peroxisomal β-oxidation[9,11]. Based on these alterations, ACBD5 deficiency was initially recognized as a peroxisomal single enzyme deficiency caused by the loss of the protein's acyl-CoA binding function potentially required for the recruitment of activated fatty acids from the cytosol for their import into peroxisomes. Our recent findings point to a role of ACBD5-VAP-mediated membrane contacts in organelle positioning and mobility as well as the transfer of ER-synthesized phospholipids to peroxisomes, where they are required for membrane expansion[5,6]. Moreover, the ER and peroxisomes cooperate in ether phospholipid and docosahexaenoic acid synthesis pathways. Therefore, peroxisome-ER membrane contacts might facilitate the exchange of lipid intermediates between both organelles[12]. Accordingly, the loss in ACBD5 might lead to a disease pathology combining symptoms of disrupted acyl-CoA acceptor and tethering protein functions. Animal models can increase our understanding on the role of peroxisome-ER contact sites in a potentially unique pathologic mechanism in ACBD5 deficiency. The C57BL/6N-A$^{tm1Brd}$ Acbd5$^{tm1a(EUCOMM)Wtsi/WtsiCnbc}$ mouse line (in the following termed Acbd5$^{-/-}$) was used to characterize pathologic alterations at the organ, cellular, molecular, and metabolic level. Like human patients, the Acbd5$^{-/-}$ mouse exhibits moderately elevated VLCFA and develops a progressive locomotor phenotype mirrored by pathologic cerebellar alterations. Remarkably, an extensive reduction in peroxisome-ER membrane contacts could be verified in hepatocytes of the mice using serial section scanning electron microscopy (S³EM), implying that a loss in the protein's tethering function contributes to the pathologic phenotype. In line, capacities for peroxisome membrane extension were reduced under peroxisome proliferating conditions. Most importantly, lipidomic profiling of cerebellum and liver revealed profound, tissue-specific alterations in the FA composition of major lipid classes implying a dysregulation of intracellular lipid homeostasis beyond a mere β-oxidation defect in ACBD5-deficient mice.

## Results

### ACBD5-loss impedes peroxisomal membrane expansion under peroxisome proliferating conditions

Acbd5$^{-/-}$ littermates exhibit an efficient block in ACBD5 expression as shown by qRT-PCR of mRNA from different tissues (n = 6 mice, three replications each) and immunodetection in mouse embryonic fibroblasts (MEF) isolated from the Acbd5$^{-/-}$ strain (Fig. 1a, c). ACBD4 is a second acyl-CoA binding protein promoting peroxisome-ER associations by VAPB interaction[4,13]. Thus, ACBD5 loss might be

compensated by ACBD4. However, qPCR data for the peroxisomal ACBD4 isoform 2 revealed that although ACBD4 is found in the same tissues as ACBD5, its gene expression did not significantly change to compensate the lack in ACBD5 (Fig. 1b).

In line with observations from human ACBD5-deficient patient fibroblasts[9,11], Acbd5$^{-/-}$ and Acbd5$^{+/+}$ MEFs exhibit comparable peroxisome numbers implying a functional peroxisome biogenesis. While Acbd5$^{+/+}$ MEFs contain heterogeneous peroxisomes with a spherical and elongated morphology, elongated peroxisomes are scarce in Acbd5$^{-/-}$ MEFs (Fig. 1c, d). Peroxisomal matrix (ACOX1, catalase) and membrane proteins (PEX3, ABCD3, PEX14) show a high degree of colocalisation in both strains, indicating no obvious defects in peroxisome maturation from either pre-peroxisomes or during asymmetric growth and division (Supplementary Fig. 1). Knockdown of ACBD5 decreased peroxisomal membrane expansion capacities in DRP1- and MFF-deficient fibroblasts, suggesting a compromised phospholipid transfer from the ER to peroxisomes in response to disrupted organelle membrane contacts[5,6]. Thus, the lack of elongated peroxisomes in Acbd5$^{-/-}$ MEFs might still indicate differences in the process of peroxisome formation. To analyze if the lack of ACBD5 reduces peroxisomal membrane expansion, we monitored peroxisome morphology under conditions inducing peroxisome elongation in the MEFs. Peroxisome elongation—but not proliferation—can be induced by docosahexaenoic acid (DHA)[14]. In standard culture medium, both Acbd5$^{-/-}$ as well as Acbd5$^{+/+}$ MEFs contain largely equal peroxisome numbers (Fig. 1e, Supplementary Fig. 1). Incubation of both MEFs with 150 µM DHA did not alter total peroxisome numbers, corroborating results published for human fibroblasts[14]. However, increased quantities of elongated peroxisomes were observed in Acbd5$^{+/+}$ but not in Acbd5$^{-/-}$ MEFs after 12 h of DHA treatment. To quantify peroxisome elongation, total peroxisome area per cell as well as average size and circularity for individual peroxisomes were determined (Fig. 1f–h). No measurable effect of DHA was observed in Acbd5$^{-/-}$ MEFs, while Acbd5$^{+/+}$ MEFs responded with elevations in average area/peroxisome as well as total peroxisome area per cell. Additionally, the difference in the average peroxisome circularity index between Acbd5$^{+/+}$ and Acbd5$^{-/-}$ MEFs increased (1.0 = round particle) after DHA incubation, further confirming peroxisome elongation in Acbd5$^{+/+}$ but not Acbd5$^{-/-}$ MEFs. These results imply that while peroxisome maintenance is guaranteed in Acbd5$^{-/-}$ MEFs, peroxisomes elongate less efficiently under conditions inducing peroxisomal membrane expansion. Thus, peroxisome plasticity appears to be reduced in Acbd5$^{-/-}$ MEFs supporting the proposed role of ACBD5-mediated peroxisome-ER contacts in membrane lipid transfer.

### Acbd5$^{-/-}$ mice develop a progressive degenerative phenotype in the cerebellum

Breeding of heterozygous Acbd5$^{+/-}$ mice produced offspring of normal habitus at Mendelian distribution of 24.7% (Acbd5$^{+/+}$), 50.6% (Acbd5$^{+/-}$), and 24.7% (Acbd5$^{-/-}$) (n = 97), indicating unreduced intrauterine survival rates. At birth homozygous Acbd5$^{-/-}$ mice are indiscernible from age-matched Acbd5$^{+/+}$ animals, equally gain size and weight and are fertile (Fig. 2c). To examine potential metabolic alterations, mitochondrial and peroxisomal lipid metabolites were measured in plasma and tissues (age 3 months). Representing mitochondrial β-oxidation, saturated and unsaturated acylcarnitines with a chain length of C2–C18 exhibited no significant differences between Acbd5$^{+/+}$ and Acbd5$^{-/-}$ mice (see Supplementary data). For peroxisomal metabolism, C22:0–C26:0 fatty acids and phytanic acid were measured[10]. While no differences for phytanic, pristanic, and behenic acid (C22:0) could be observed, Acbd5$^{-/-}$

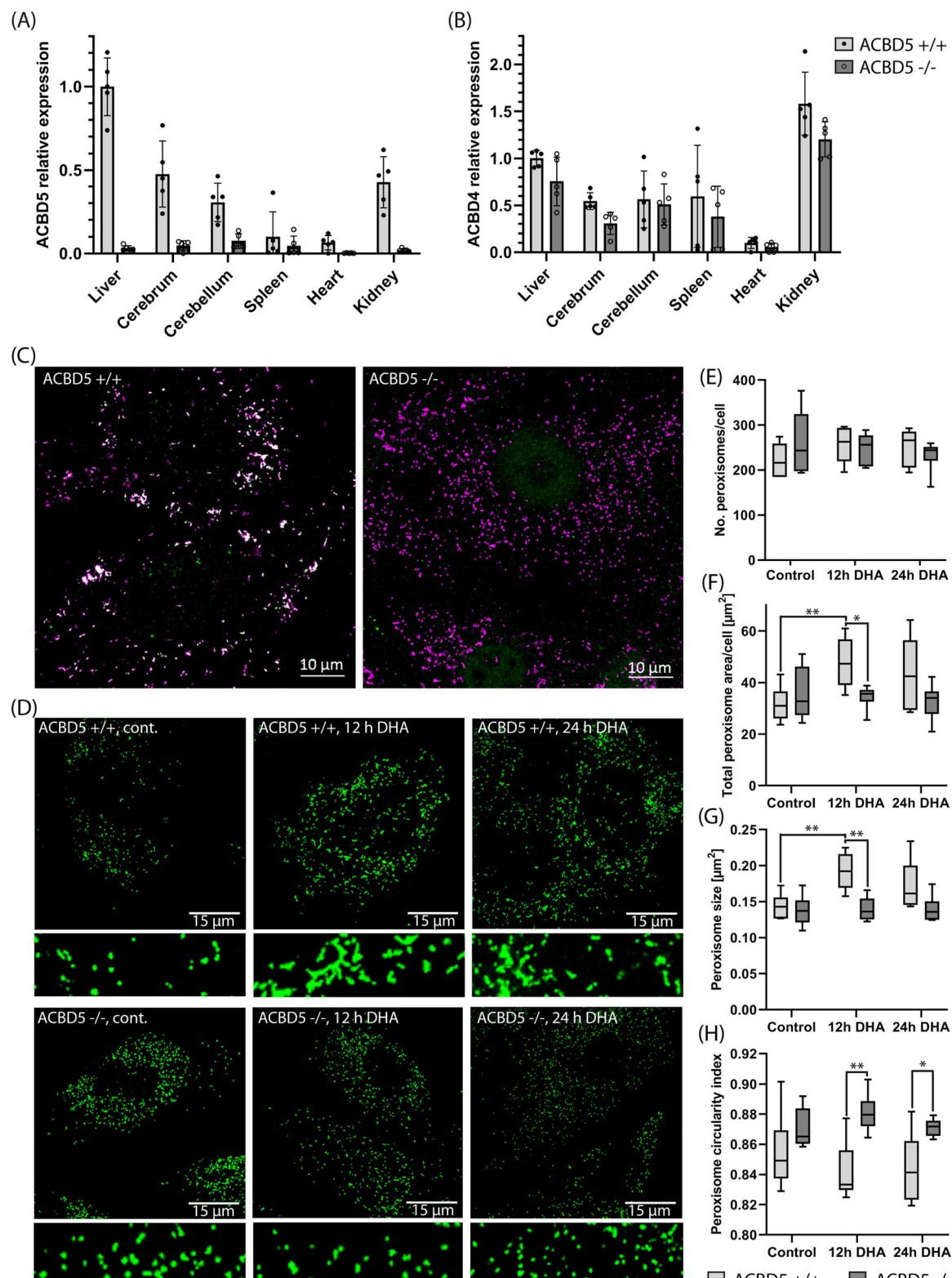

**Fig. 1 Confirmation of ACBD5 knockout and effects on peroxisome morphology in MEF. a** Expression of ACBD5 in selected tissues of *Acbd5*$^{+/+}$ and *Acbd5*$^{−/−}$ mouse strains as determined by RT-qPCR. GAPDH expression was measured for calculation of the ΔCT values. Relative expression in the tissues was normalized to *Acbd5*$^{+/+}$ liver tissue (set to 1.0). **b** Expression of ACBD4 determined as above. **c** Antibody staining for ACBD5 in MEFs from *Acbd5*$^{+/+}$ and *Acbd5*$^{−/−}$ mouse strains (green). PEX14 was labeled as a marker for peroxisomes (magenta). Bar graphs show data means with standard deviations. **d** Peroxisomes (antibody: PEX14) in MEF cells after 12 h and 24 h incubation with DHA. Narrow cut-outs represent ×4 magnifications from the images placed directly above. **e** Quantification of cellular peroxisome number, (**f**) total peroxisome area covered per cell, (**g**) average particle size, and (**h**) average index of circularity in MEFs incubated in 150 μm DHA. Box plots depict the interquartile range, medians, and minima maxima (No. of experiments = 6250 cells were quantified in each group, *$p < 0.05$, **$p < 0.01$, ***$p < 0.001$; unpaired *t*-test, two-sided).

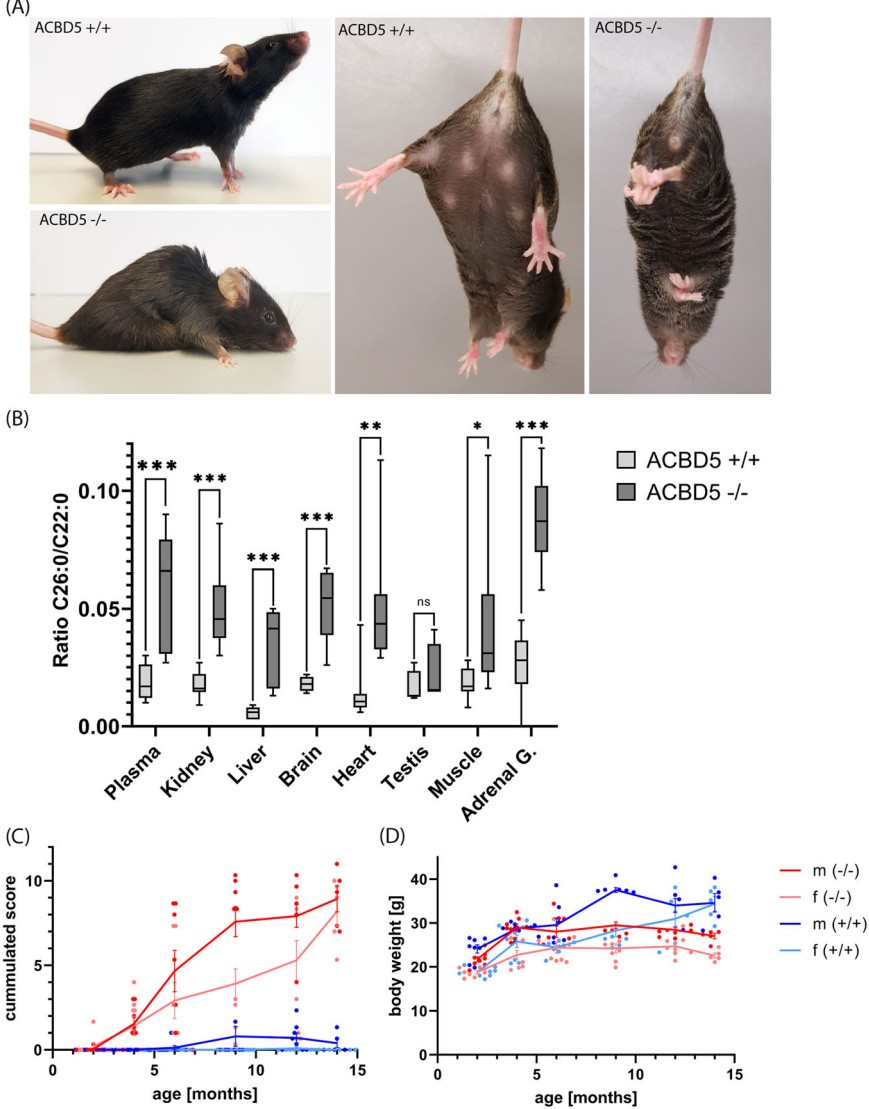

**Fig. 2 General morphology of *Acbd5*$^{-/-}$ mice. a** Habitus of 1-year-old *Acbd5*$^{-/-}$ mice—the animals show examples for kyphosis and hind limb clasping observed in this study. **b** C26:0/C22:0 ratios of acyl-CoA concentrations in plasma and tissues of 3 months aged *Acbd5*$^{+/+}$ and *Acbd5*$^{-/-}$ mice ($n = 5$/gender). **c** Average body weight of *Acbd5*$^{-/-}$ and *Acbd5*$^{+/+}$ male and female mice ($n = 8$ per gender and age). **d** Composite phenotype scores for qualitative tests for hind limb clasping, ledge test, gait, and kyphosis observed in male and female *Acbd5*$^{-/-}$ and *Acbd5*$^{+/+}$ mice ($n = 8$ per gender and age). Each test was separately scored using values from 0 (not affected) to 3 (severely affected) with a combined score up to 12 for all four measures[15]. Box plots depict the interquartile range, medians, and minima/maxima, line graphs show means with standard deviations (*$p < 0.05$, **$p < 0.01$, ***$p < 0.001$; unpaired *t*-test, two-sided).

mice exhibited slightly elevated levels of lignoceric acid (C24:0) and three-fold elevations in cerotic acid (C26:0) in plasma and tissues (Fig. 2, Supplementary Fig. 2). Erythrocyte plasmalogen levels, representing peroxisome ether lipid synthesis, showed no significant differences between *Acbd5*$^{-/-}$ and *Acbd5*$^{+/+}$ mice (see Supplementary data). Likewise, urinary metabolites representing a defect in the peroxisomal part of the bile acid synthesis pathway were unaltered (Supplementary Fig. 2). Thus, *Acbd5*$^{-/-}$ mice present at first sight with typical but rather moderate lipid alterations of a peroxisomal β-oxidation disorder. While *Acbd5*$^{-/-}$ mice appear to be unaffected during adolescence, they develop a striking kyphosis and hind limb clasping, which are both signs for degenerative processes in the cerebellum (Fig. 2a). These phenomena were accompanied by the development of an unsteady gait and problems in movement coordination (Supplementary Movie 1, 2). To evaluate the time-scale of the ataxia development, a scoring system consisting of four simple

behavioral tests—hind limb clasping, gait performance, degree of kyphosis, ability to balance on ledge—was applied[15]. For each test, the phenotype degree was rated between 0 and 3. While *Acbd5*$^{+/+}$ mice did not show summated scores above 2, scores of *Acbd5*$^{-/-}$ mice increased progressively from the age of 4 months to the end of the testing period at 14 months (Fig. 2d). As reflected by the standard deviations, disease severity varies to some extent among age-matched individuals. Both male and female *Acbd5*$^{-/-}$ mice above 6 months showed lower weights and food consumption rates if compared to controls (Fig. 2c, Supplementary Fig. 2), suggesting that the locomotor deficiency impeded food intake from the containers at the cage top. Easier access to the food at the cage floor; however, increased food consumption comparably in both strains indicating that the locomotor deficits might not be the only reason for the weight differences (Supplementary Fig. 2). Moreover, successful mating was not observed with elder breeding pairs (>4 months).

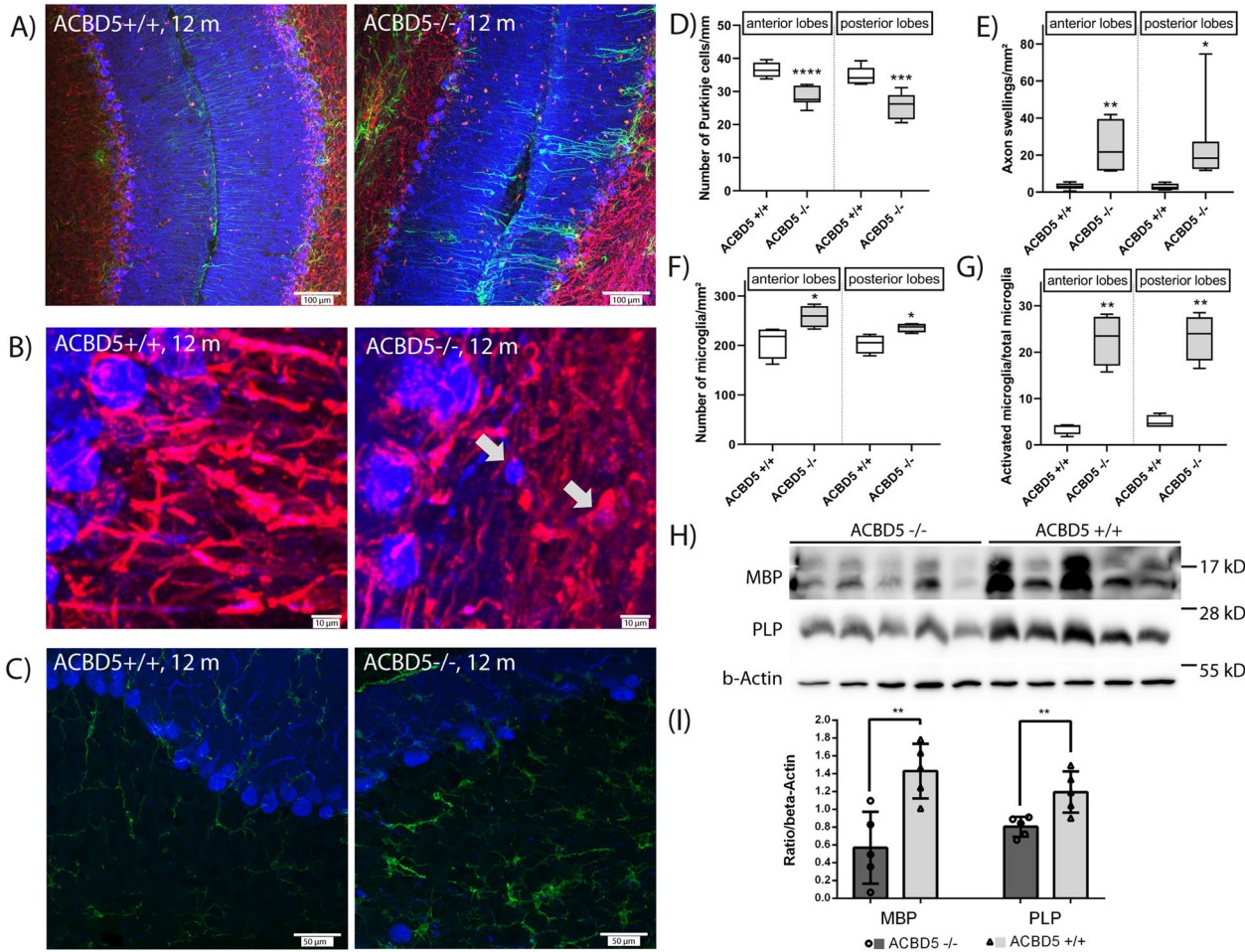

**Fig. 3 Cerebellar phenotype observed in 1-year-old male *Acbd5*$^{-/-}$ mice. a** Visualization of Purkinje cells (calbindin, blue), astro- (GFAP, green) and oligodendroglia (MBP, red) in cerebelli of 1-year-old male *Acbd5*$^{-/-}$ and *Acbd5*$^{+/+}$ mice. **b** Magnification of the granular layer from the overviews depicted in (**a**). Arrows highlight swellings in Purkinje cells axons. Note that axonal swellings were either found still surrounded or without a myelin layer as indicated by the MBP signal (red). **c** Overviews of microglia in the cerebelli of *Acbd5*$^{-/-}$ and *Acbd5*$^{+/+}$ mice; microglia was stained with antibodies against IBA-1 (green); for orientation, Purkinje cells are visualized by calbindin antibodies (blue). Note the presence of activated, ameboid microglia in the *Acbd5*$^{-/-}$ cerebelli. **d** Quantification of Purkinje cell densities in cerebelli from 1-year-old *Acbd5*$^{-/-}$ and *Acbd5*$^{+/+}$ males. ($n = 7$, in average 12 mm Purkinje cell layer/animal). **e** Quantification of axon swelling found in Purkinje cell axons of the granular layer ($n = 7$, quantification of in average 2 mm$^2$ granular layer/animal). **f, g** Quantification of total number and number of amoeboid microglia in the cerebellar cortex of 1-year-old *Acbd5*$^{-/-}$ and *Acbd5*$^{+/+}$ ($n = 4$; quantification of in average 4 mm$^2$ of cerebellar cortex/animal). **h** Immunoblots against MBP and PLP to estimate axon myelinization in the cerebellum (full size immunoblots are shown in Supplementary Fig. 7). Actin was applied as a loading control and used for normalization of the (**i**) MBP and PLP intensity quantification. All box plots depict the interquartile, medians, and minima/maxima, all bar graphs data means with standard deviations (*$p < 0.05$, **$p < 0.01$, ***$p < 0.001$; unpaired *t*-test, two-sided).

The cerebellum plays an important role in limb and eye movement coordination, balance, and walking. As the locomotor deficits, kyphosis, and hind limb clasping of *Acbd5*$^{-/-}$ mice suggested a potential cerebellar contribution to the disease pathology, sagittal sections of cerebelli from 1-year-old *Acbd5*$^{-/-}$ males—a stage with a pronounced locomotor phenotype—were used to evaluate morphological alterations by immunofluorescence microscopy. Purkinje cell numbers were significantly reduced by 22 and 26% in cerebellar anterior lobes 3–6 and posterior lobes 7–9, respectively (Fig. 3a, d). Additionally, Purkinje cell axons regularly exhibited swellings as a typical indicator for intracellular transport defects during axon degeneration[16] (Fig. 3b). Whereas axon swellings were rarely found in anterior and posterior lobes of *Acbd5*$^{+/+}$ cerebellar granular layers (3 swellings/mm$^2$), significantly higher amounts of 24 ($p < 0.01$) and 26 swellings/mm$^2$ ($p < 0.05$) were identified in *Acbd5*$^{-/-}$ mice (Fig. 3e). Number, distribution, and morphology

of peroxisomes in the soma of Purkinje cells, however, were largely unchanged (Supplementary Fig. 3A–C). At advanced stages, many peroxisomal disorders exhibit axon demyelination, which was not clearly obvious in immunofluorescence images against MBP (Supplementary Fig. 3D). Myelin basic protein (MBP) and proteolipid protein (PLP) intensities in immunoblots of cerebellar homogenates of *Acbd5*$^{+/+}$ and *Acbd5*$^{-/-}$ mice (age 1 year), however, indicate a slight but significant demyelination (Fig. 3h, Supplementary Fig. 4D, E).

In many CNS pathologies, astrocytes become activated in a process termed reactive (astro)gliosis[17]. Major hallmarks are a hypertrophy of astrocytic processes and the increased expression of the intermediate filament component glial fibrillary acidic protein (GFAP). GFAP staining in cerebellar sections revealed that especially radial-type Bergmann glia, which expressed GFAP in low level in controls, underwent reactive gliosis characterized by intense GFAP staining of their radial processes (Fig. 3a). Of

note, Bergmann glial cells showed a strong reaction in places with missing Purkinje cells. Likewise, microglia frequently transforms into an activated state during cerebellar ataxia[18]. Activation leads to increases in microglial density, ramification, and elevated expression of ionized calcium-binding adapter molecule 1 (IBA1) ultimately triggering neuroinflammation[19]. IBA1 immunostaining revealed an elevation in ramified microglia from 5 to 23% in cerebellar lobes of $Acbd5^{-/-}$ mice, whilst microglia numbers increased only moderately, indicating a mild neuroinflammation (Fig. 3c, f, g).

To assess if the degenerative processes affect synaptic communication between Purkinje cells and incoming excitatory parallel and climbing fibers or inhibitory interneurons, we assessed signal densities of the vesicular glutamate receptors VGLUT1, VGLUT2 and the vesicular inhibitory amino acid transporter VGAT. To avoid that potential alterations in synapse numbers would merely mirror the absence of degenerated Purkinje cells, the quantification was restricted to areas with still continuous Purkinje cells. For all three synapse types, no significant alterations in density were detected indicating that innervation is unaffected in the early phases of Purkinje cell degeneration (Fig. 4a, b). During postnatal development of the cerebellum, synapses of climbing fibers change from a perisomatic to a more distal localization in the molecular layer[20]. As visualized by the VGLUT2 staining (Fig. 3a), distribution of climbing fiber synapses inside the molecular layer was unaltered in $Acbd5^{-/-}$ mice, implying that developmental cerebellar wiring is intact. Therefore, an early onset of the pathologic alterations during brain development is unlikely.

In summary, moderate but significant morphological changes in the cerebellum of $Acbd5^{-/-}$ mice indicate that a progressive cerebellar degeneration accompanied by a mild neuroinflammation contributes to the animal's locomotor deficits. It should be highlighted, that in addition to the degeneration in the cerebellum, pathologic processes in other brain motor centers could contribute to the locomotor phenotype (e.g. basal ganglia, motor cortex, and pyramidal tracts); corresponding investigations will be performed in the future. Of note, resembling human patients[8], a retinal degeneration characterized by reduced photoreceptor cells, increase in microglia and astrocyte activation (Supplementary Fig. 4A–C) points to a more widespread pathology in the CNS of $Acbd5^{-/-}$ mice.

To investigate if the Purkinje cell degeneration is preceded by defects in their axons, we analyzed axon initial segment (AIS) morphology (Fig. 4c). The AIS separates the more distal axon part from the somatodendritic compartment and is the site of action potential generation[21]. AIS integrity is compromised in experimental models of neuroinflammatory and demyelinating disorders[22–24]. Moreover, alterations in AIS length and positioning affect the excitability and firing behavior of neurons and can reflect changes in neuronal activity[21,25]. A complete AIS loss as reported for experimental drug-induced models for neuroinflammation and demyelination[22,23] was not observed for $Acbd5^{-/-}$ Purkinje cells. Also, AIS length did not differ between $Acbd5^{+/+}$ and $Acbd5^{-/-}$ mice, whilst the distance between soma and the AIS was slightly but significantly increased in the latter (0.52 vs 0.79 μm) (Fig. 4d). Changes in AIS positioning in the same range were reported in a mouse model for genetic epilepsy[26]. Further electrophysiological data are required to validate if such subtle changes in AIS positioning reflect physiologically relevant changes in neuronal network plasticity.

**The loss of ACBD5 is accompanied by elevated peroxisomes numbers in hepatocytes.** Associated with the prominent peroxisomal liver metabolism, various peroxisome disorders exhibit a liver phenotype, which can include steatosis, fibrosis, hepatomegaly, or cholestasis[27]. Macroscopically, $Acbd5^{-/-}$ livers did not reveal obvious alterations and hepatosomatic indices remained comparable to $Acbd5^{+/+}$ mice (0.038 ± 0.003 vs.0.041 ± 0.003 males, 3 months; 0.040 ± 0.004 vs 0.043 ± 0.003 females, 3 months; 0.030 ± 0.06 vs 0.029 ± 0.05 males, 1 year; 0.032 ± 0.05 vs 0.028 ± 0.04 females, 1 year). To assess a potential liver pathology, sections of 3-month and 15-month-old male $Acbd5^{+/+}$ and $Acbd5^{-/-}$ mice were subjected to HE and oil-red staining (Fig. 5a, b). According to the HE staining, liver morphology was unchanged in $Acbd5^{-/-}$ mice at both ages (Fig. 5a). Oil-red staining showed numerous miniature lipid droplets (LD) in both animal strains. Differences, however, were observed for LD with a diameter >2 μm (Fig. 5b, d). Such LD were significantly increased in livers of $Acbd5^{-/-}$ mice aged 3 months, suggesting that accumulating VLCFA are deposited in LD as triacylgycerols (TG). The 15-month-old $Acbd5^{-/-}$ mice showed, however, significantly reduced amounts of such large LD (Fig. 5b, d). In line lipidomics data show a tendency to reduced liver TG levels (Supplementary Fig. 6A). The lower LD numbers are in line with the lower body weight of older $Acbd5^{-/-}$ mice, pointing to a generally reduced fat storage.

Elevated VLCFA can induce PPARα-mediated peroxisome proliferation as observed for ACOX1-deficient mice[28]. Thus, VLCFA elevations in $Acbd5^{-/-}$ mice may induce peroxisome proliferation in hepatocytes (Fig. 2b). PEX14 and catalase immunostaining revealed that peroxisomes cover larger areas of the hepatocytes in $Acbd5^{-/-}$ mice aged 3 and 15 months than in controls (Fig. 5c, e, Supplementary Fig. 5A). Livers of 3-month-old mice were additionally analyzed at the ultrastructural level (n = 3). Quantification of peroxisome numbers revealed a 3.3-fold increase in $Acbd5^{-/-}$ hepatocytes, corroborating the immunofluorescence analysis (Fig. 6a, b). As in MEF, peroxisomes from $Acbd5^{-/-}$ hepatocytes showed a tendency to more circular forms (Supplementary Fig. 5B, C). By contrast, the ultrastructure of mitochondria and glycogen content were unchanged.

PPARα-mediated peroxisome proliferation is accompanied by the induction of peroxisomal β-oxidation as well as mitochondrial and ER lipid metabolizing enzymes[29]. To gain further insight into the nature of peroxisome proliferation in $Acbd5^{-/-}$ mice, the relative abundance of selected lipid metabolic proteins was analyzed in liver post-nuclear supernatants (PNS, Fig. 6c, d) and purified peroxisomes (Fig. 6e). In the PNS of $Acbd5^{-/-}$ mice, the peroxisomal proteins participating in FA degradation, ACOX1, LBP, ABCD3 as well as the multilocalized acyl-CoA synthase ACSL4 (peroxisomes, ER, mitochondria) exhibited increased abundance (Fig. 6c, d) in line with elevated ACOX enzymatic activities (factor 2.2) (Fig. 6f). The rate-determining enzyme of mitochondrial β-oxidation CPT1 and the ER-resident FA elongase ELOVL1 are also induced by PPARα[30,31]. Levels of CPT1 and ELOVL1, however, were not significantly increased in the PNS of $Acbd5^{-/-}$ livers. Isolated peroxisomes from rodents treated with the PPARα agonist bezafibrate show a marked increase in ABCD3, LBP and to a lesser extent ACOX1 [2]. In isolated peroxisomes from $Acbd5^{-/-}$ livers; however, ABCD3, ACOX1, and LBP were unaltered like PPARα-independent peroxisomal proteins (PEX14, catalase) (Fig. 6e, columns G4 + G5. Supplementary Fig. 5D). Corroborating the ACOX1 immunoblot signals, isolated, lysed peroxisome fractions from $Acbd5^{-/-}$ and $Acbd5^{+/+}$ mice exhibited comparable ACOX activities (Fig. 6f). The data imply that either moderately elevated VLCFA trigger a weak PPARα-mediated peroxisome proliferation with no major changes in the peroxisomal proteome composition or a peroxisome proliferation through a PPARα-independent mechanism.

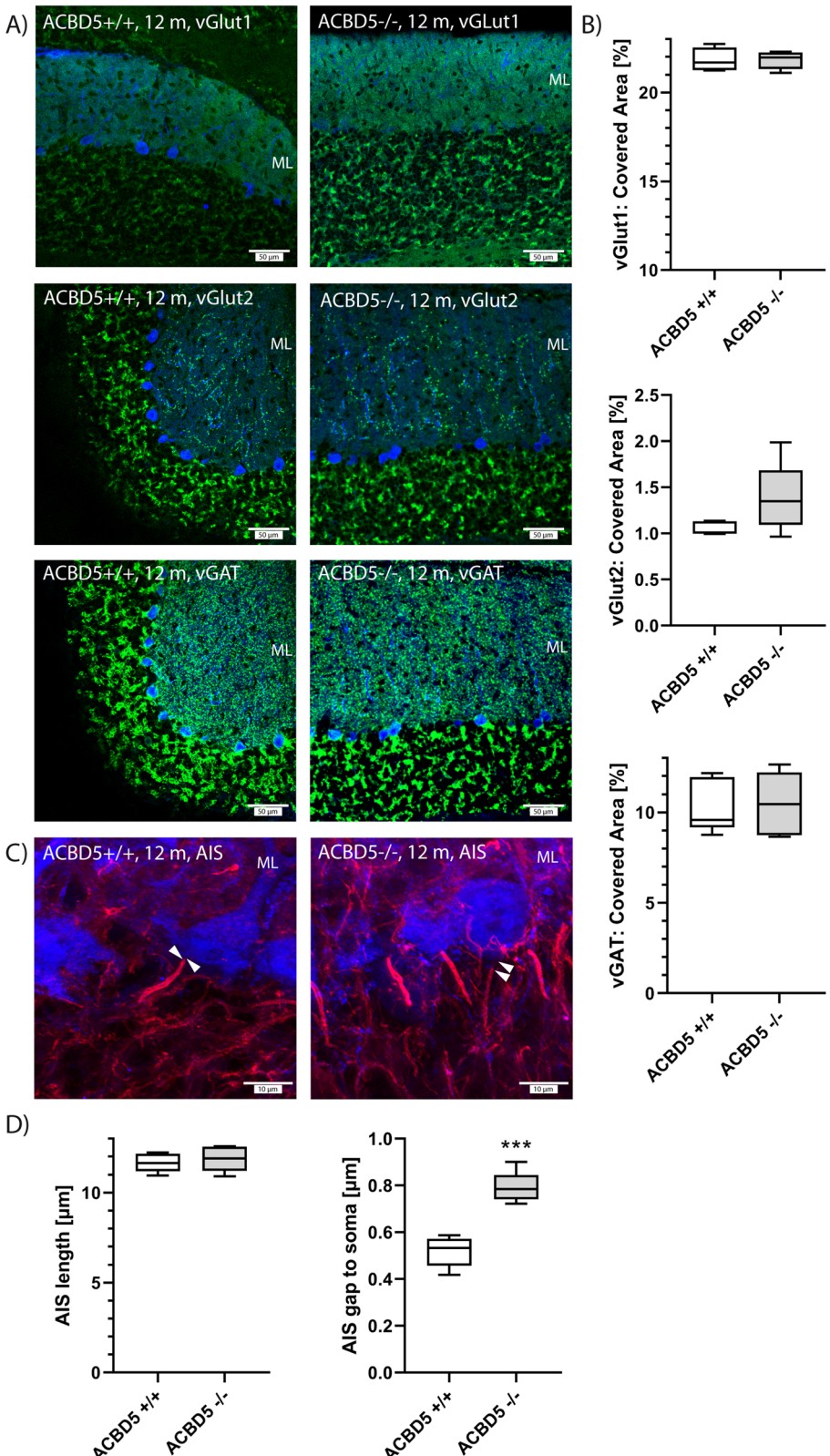

**Peroxisome-ER contact sites are highly reduced in $Acbd5^{-/-}$ hepatocytes**. In addition to its acyl-CoA binding properties, ACBD5 facilitates membrane contacts with the ER via interaction with VAPB[5,6]. Thus, it is obligatory to analyze if the lack of ACBD5 reduces peroxisome-ER contacts in vivo or if other tethering complexes compensate its loss. Peroxisome-ER contact quantification in cell lines revealed that approximately 65–85% of peroxisomes are associated with ER membranes[5,32]. For hepatocytes, we observed a value of 82% (membrane distance peroxisome-ER < 30 nm) in $Acbd5^{+/+}$ mice (Fig. 6i). Remarkably, only 10% of peroxisome-ER associations were found in $Acbd5^{-/-}$ hepatocytes, while ER tubules occasionally surrounded peroxisomes at higher distances. Membrane associations between peroxisomes and mitochondria, which occurred in a frequency of

**Fig. 4 Status of excitatory and inhibitory synapses as well as the AIS of *Acbd5*⁻/⁻ Purkinje cells. a** Images visualizing distribution of excitatory (VGLUT1, VGLUT2) and inhibitory synapses (VGAT) in the cerebellar molecular layer (green). For orientation, Purkinje cells are marked by calbindin antibodies (blue); VGAT and VGLUT2 were labeled in parallel using different fluorophore-coupled secondary antibodies but for clarity reasons are both shown in separate images. **b** Quantification of fluorescent signal densities for VGLUT1, VGLUT2, and VGAT in the cerebellar molecular layer. ($n = 5$, in average 0.13 mm² molecular layer/animal). **c** Colocalization of calbindin (blue) and ankyrinG (red) antibody signals were used to define the AIS in Purkinje cells. The gap between an AIS (red) and correspondent Purkinje cell soma (blue) is highlighted by arrowheads in both images. **d** Quantification of length and axonal position of the Purkinje cell AIS ($n = 5$, >50 AIS/animal). For orientation, the cerebellar molecular layer is indicated by ML in all images (*$p < 0.05$, **$p < 0.01$, ***$p < 0.001$; unpaired $t$-test, two-sided; whisker plots depict the interquartile range, medians, and minima/maxima).

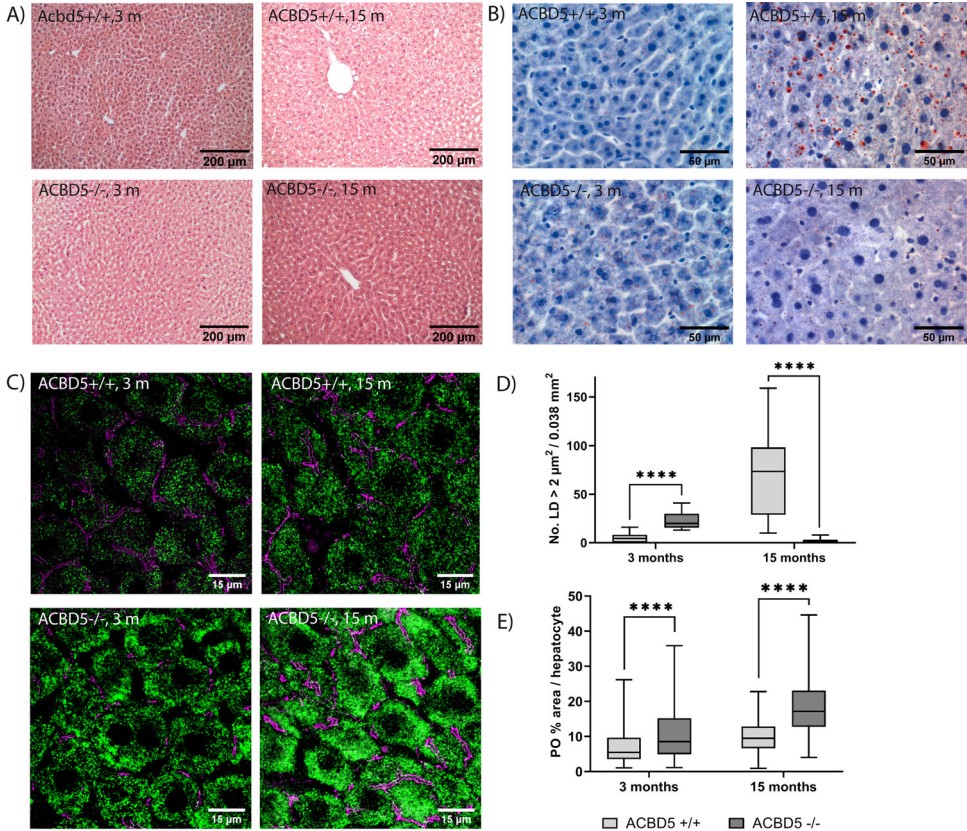

**Fig. 5 Liver phenotype in male *Acbd5*⁻/⁻ mice aged 3 and 15 months. a** Representative HE staining of livers from *Acbd5*⁺/⁺ and *Acbd5*⁻/⁻ mice. **b** Representative images of lipid droplets visualized by oil-red staining in livers from *Acbd5*⁺/⁺ and *Acbd5*⁻/⁻ mice. **c** Peroxisomes in hepatocytes from *Acbd5*⁺/⁺ and *Acbd5*⁻/⁻ mice; peroxisomes are visualized by Pex14 antibody staining (green), antibodies against ZO1 (tight junctions, magenta) were used to outline hepatocyte cell boundaries. **d** Quantification of lipid droplets (>2 μm) in livers from *Acbd5*⁺/⁺ and *Acbd5*⁻/⁻ mice ($n = 5$, quantification based on 0.075 mm²/liver). **e** Quantification of peroxisome abundance in hepatocytes as shown in (**c**), data were quantified as ratio of total peroxisome (PO) covered per hepatocyte ($n = 5$ animals/group, 0.0 5 mm²/liver. (*$p < 0.05$, **$p < 0.01$, ***$p < 0.001$, ****$p < 0.0001$; unpaired $t$-test, two-sided; all box plots depict the interquartile range, medians and minima/maxima).

3%, did not deviate between both mouse strains. However, peroxisomes in triple associations connecting peroxisomes to mitochondria via an interjacent ER tubule did decrease from 27.5% to 2.8% (Fig. 6i). Remarkably, peroxisomes in *Acbd5*⁻/⁻ hepatocytes showed more heterogeneous shapes than in *Acbd5*⁺/⁺ controls and often localized in clusters (Fig. 6a). Since two-dimensional EM images do not allow to visualize complete organelles, serial ultrathin sections were analyzed to reconstruct the form and extension of peroxisomes as well as their positioning to other organelles using S₃EM[33]. Remarkably, *Acbd5*⁻/⁻ hepatocytes contained large clusters of multiple spherical to oval-shaped peroxisomes devoid of other organelles (Fig. 6h). Of note, clustered peroxisomes do not form extended tubules or reticular networks. In the vicinity, mitochondria were frequently surrounded by sheet-like ER structures underlining a subcellular separation between the peroxisomes and remaining organelles. In

*Acbd5*⁺/⁺ cells, by contrast, peroxisomes are distributed as single or small groups of organelles which are individually surrounded by ER tubules. Average minimum distances between peroxisome and ER exhibited a value of 25 nm for *Acbd5*⁺/⁺ and 106 nm for *Acbd5*⁻/⁻ hepatocytes (Fig. 6j). Notably, tethering pairs can span considerable distances between the opposing membranes[1]. Hence, these findings might indicate a preserved "high distance" tethering complex in *Acbd5*⁻/⁻ mice.

Despite the only moderately elevated peroxisome numbers observed by microscopy, total protein yields for isolated peroxisomes from *Acbd5*⁻/⁻ livers by far exceeded those from controls (average factor of 17) (Fig. 6g). Notably, VAPB, which is most abundant in ER-enriched fractions, is more intense in the peroxisome fractions from *Acbd5*⁺/⁺ than *Acbd5*⁻/⁻ mice (Fig. 6e, columns G4, G5, Supplementary Fig. 5D). Therefore, ER membranes, which can still be attached to isolated

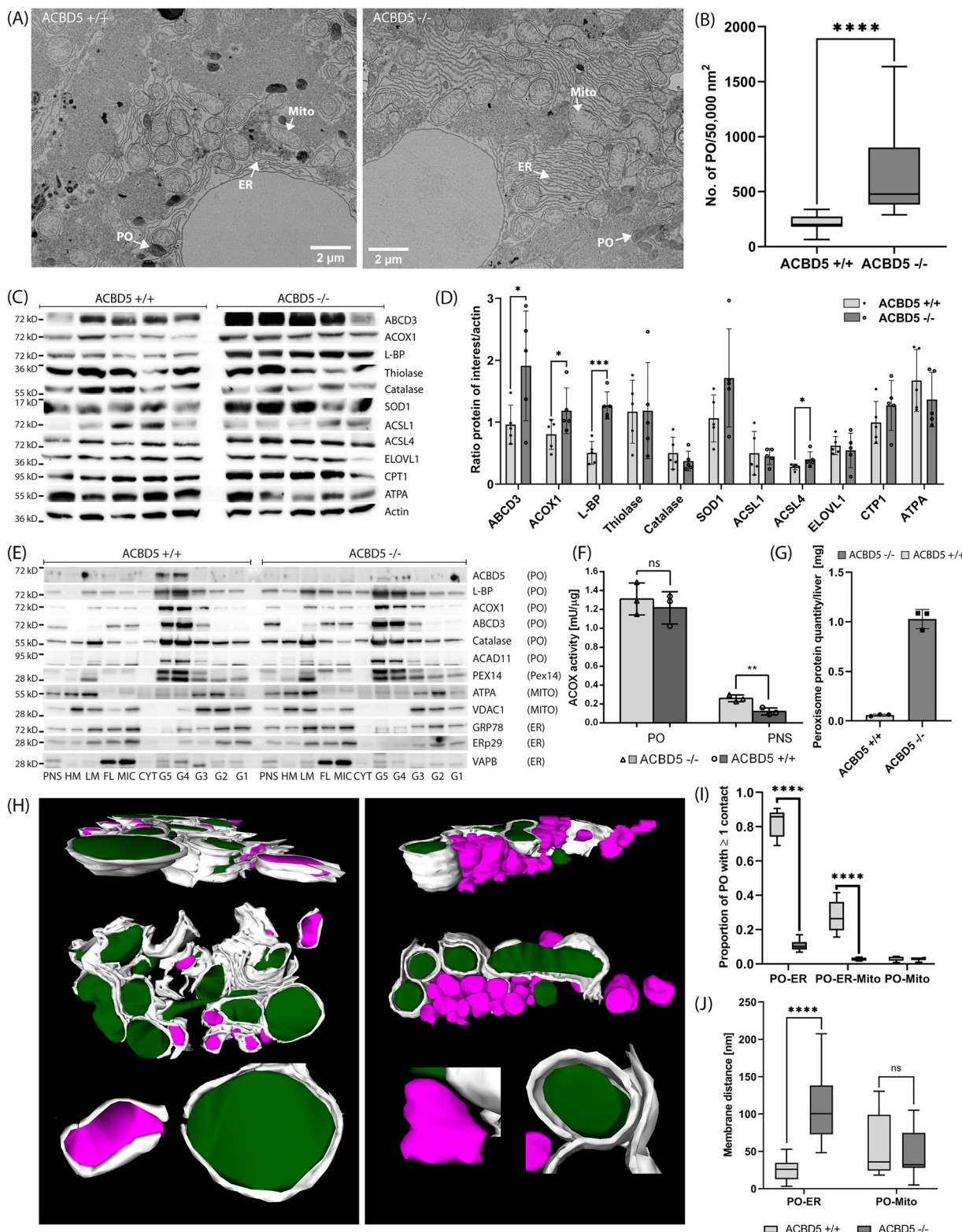

peroxisomes[34], appear to be less abundant in peroxisome fractions from *Acbd5*[−/−] mice. Thus, disruption of peroxisome-ER contacts may facilitate the migration of a higher peroxisome proportion into the gradient's high-density regions allowing a more efficient separation between peroxisomes and microsomes. Likewise, increasing ACBD5 and VAPB expression in HepG2 cells induced a shift of peroxisome density towards the gradient's lower density region[5].

In summary, the loss of ACBD5 disrupts close proximity ER contacts and induces the formation of large peroxisome clusters excluding other organelles. Overall similar peroxisome sizes (Supplementary Fig. 5B, C), comparable ACOX activities, unchanged amounts of peroxisomal matrix and membrane proteins (Fig. 6) and preserved organelle densities of isolated peroxisomes show that the drastic reduction in ER contacts supposedly transmitting membrane lipids for peroxisome

**Fig. 6 Organelle and protein alterations in the liver of *Acbd5*⁻/⁻ mice. a** Representative EM images of hepatocytes from *Acbd5*⁺/⁺ (WT) and *Acbd5*⁻/⁻ mice ($n = 3$, age 3 months) with peroxisomes labeled by alkaline DAB staining. **b** Average peroxisome abundance in hepatocytes. **c** Immunoblots from liver homogenates ($n = 5$/group) detecting proteins related to pathways involved in cellular lipid metabolism. L-BP—peroxisomal L-bifunctional protein, Thiolase—peroxisomal 3-ketoacyl-CoA thiolase, ATPA—ATP synthase subunit alpha (full size immunoblots are shown in Supplementary Fig. 8). **d** Relative average intensities of the proteins detected in (**c**) after normalization against actin. **e** Peroxisomal protein abundance in subcellular liver fractions (full size immunoblots are shown in Supplementary Fig. 9). PNS—post-nuclear supernatant, HM—heavy mitochondrial fraction, LM—light mitochondrial fraction, FL—fluffy LM layer, MIC—microsomal fraction, CYT—cytosol, G—gradient fractions (density increasing with numbers), L-BP—peroxisomal L-bifunctional protein, ATPA—ATP-synthase, subunit α. Columns G1 (low density)—G5 (high density) represent fractions from the final density gradient. Note that G5 and G4 comprise highly pure peroxisome fractions (see Islinger et al. [2]). **f** Acyl-CoA oxidase activities in PNS and isolated peroxisome fractions (G4 from **e**). **g** Total protein yields in peroxisome fraction (G4 + G5) isolated from livers of *Acbd5*⁺/⁺ and *Acbd5*⁻/⁻ mice (average from five isolations). **h** Representative 3D reconstruction of serial EM sections (resolution in x, y, z-axis: 3.5 × 3.5 × 50 nm), peroxisomes are shown in magenta, mitochondria in green and ER cisternae in gray. For estimation of organelle sizes see (**a**). **i** Proportion of peroxisomes showing membrane contact sites with at least one other organelle (0.01 mm²/liver from the EM images were used for quantification). **j** Average minimum distance observed between ER cisternae and peroxisomes or mitochondria calculated from the 3D-reconstructions (all box plots depict the median, interquartile range, and minima/maxima; all bar graphs show data means with standard deviations, *$p < 0.05$, **$p < 0.01$, ***$p < 0.001$, ****$p < 0.0001$; unpaired $t$-test, two-sided).

duplication do not prevent largely normal peroxisome formation in hepatocytes. Rather, ER associations might regulate proper intracellular positioning and distribution of peroxisomes possibly required for efficient metabolite exchange or transport[7]. Notably, the peroxisome clustering observed in *Acbd5*⁻/⁻ hepatocytes supports a role of the ER in intracellular organelle positioning.

**Lipidomic profiles from *Acbd5*⁻/⁻ mice reveal organelle-specific accumulations of individual lipid species.** In *Acbd5*⁻/⁻ mice and ACBD5-deficient humans increased VLCFA concentrations can be associated with reduced peroxisomal β-oxidation capacities[9,11]. Here we analyzed if changes at the lipidome level correlate with phenotypic organ alterations. Liver and cerebellum from 1-year-old male *Acbd5*⁻/⁻ mice were subjected to lipidome analysis covering >1000 lipid species[35,36], in order to decipher organ-specific alterations in the composition of membrane and storage lipids. Since ACBD5 facilitates the formation of peroxisomes-ER membrane contacts, the data presented specifically focuses on lipid pathways requiring cooperation between both organelles. Those include (1) ether phospholipid lipid synthesis, (2) synthesis of the polyenoic FA docosahexaenoic (DHA) and docosapentaenoic acid (DPA) and (3) maintaining FA chain length equilibrium by balancing FA elongation (ER) and degradation (peroxisome)[12]. Full MS datasets are provided in the Supplementary data file—Figs. 7, 8.

(1) Ether phospholipids:
Ether phospholipid synthesis is initiated in peroxisomes, but completed at the ER. Lipid analysis after ACBD5 and VAPB knockdown in HeLa cells and ACBD5-deficient patient fibroblasts revealed moderately reduced ether lipid quantities[6,35]. In *Acbd5*⁻/⁻ cerebelli, quantities of alkylphosphatidylcholines (PC[O]) and alkylphosphatidylethanolamines (PE[O]) were somewhat reduced compared to *Acbd5*⁺/⁺ mice (Fig. 7). Plasmalogens are highly abundant in myelin. Thus, the observed myelin losses might explain the reduction in ether lipids; however, other lipid species enriched in myelin like hexylceramides were unchanged (Supplementary Fig. 6A). The diminished ether lipid concentrations might therefore indicate decreased peroxisomal LCFA-CoA recruitment by ACBD5 (required for fatty alcohol synthesis) or a reduced transfer efficiency of ether lipid precursors from peroxisomes to the ER by the loss of membrane contacts. In liver, both ether lipid classes showed largely comparable values (Fig. 7). These differences between cerebellum and liver may result from the peroxisome proliferation in hepatocytes compensating a less

efficient synthesis pathway. Importantly, alkylmonoacylglycerols (DG[O]) and alkyldiacylglycerols (TG[O]) levels were significantly increased in the liver of *Acbd5*⁻/⁻ mice potentially representing a shunting of DG[O] towards TG[O] (Fig. 7). This phenomenon was not seen in the cerebellum (Fig. 7), indicating organ-specific changes in ether lipid homeostasis. TG[O] comprise between 10–20% of neutral cellular lipids, suggesting that this largely unexplored lipid class may play a central role in ether lipid metabolism[37].

(2) Synthesis of DHA and DPA:
Peroxisomes and the ER cooperate to synthetize the PUFAs DHA and DPA. DHA and DPA precursors are elongated and desaturated at the ER producing C24:6(n-3)-CoA and C24:5(n-6)-CoA intermediates and finally shortened by peroxisomal β-oxidation yielding C22:6(n-3)-CoA and C22:5(n-6)-CoA, respectively[12]. Thus, an imbalance in peroxisomal and ER pathway cooperation might alter DHA/DPA content in major lipid classes. Lysophosphatidylcholine (22:6) (LPC) and Lyso-phosphatidylethanolamine (C22:6) (LPE) can be used to estimate DHA levels in phospholipids. Slightly decreased values for liver LPC(22:6) and LPE(22:6) may point to a reduction in DHA synthesis capacities in *Acbd5*⁻/⁻ mice (Fig. 7). Cerebellar values were, however, unchanged. For comparison, LPC(20:5) and LPE(20:5) representing eicosapentanoic acid—a DHA precursor entirely synthesized at the ER[38] – were consistently elevated in both tissues. Notably, there is a shift in double bonds numbers in LPC containing VLCFA (C > 22) and ULCFA (C > 32) in *Acbd5*⁻/⁻ cerebelli. While VLCFA in LPC contains on average 3-4 double bonds, ULCFA possess preferentially 4–6 double bonds. These results may indicate that precursors of DPA and DHA instead of being rapidly imported into peroxisomes for chain-shortening accumulate at the ER, where they undergo further elongation and desaturation towards polyenoic ULCFA.

(3) Homeostasis of VLCFA
Disruptions in VLCFA catabolism as observed in *Acbd5*⁻/⁻ mice will likely impact FA composition in complex lipids. FA species distribution in LPC can be used to correlate the abundance of individual FA in phospholipids. Saturated as well as unsaturated FA with a chain length >C24 were significantly increased in LPC of *Acbd5*⁻/⁻ livers and cerebelli (Fig. 8). Only the cerebellum, however, exhibited elevated levels of highly unsaturated ultra-long chain FA (>C32:3, ULCFA). Likewise, polyenoic FA (C22-C32) possessing >3 double bonds were only elevated in cerebellar LPC (Fig. 8).

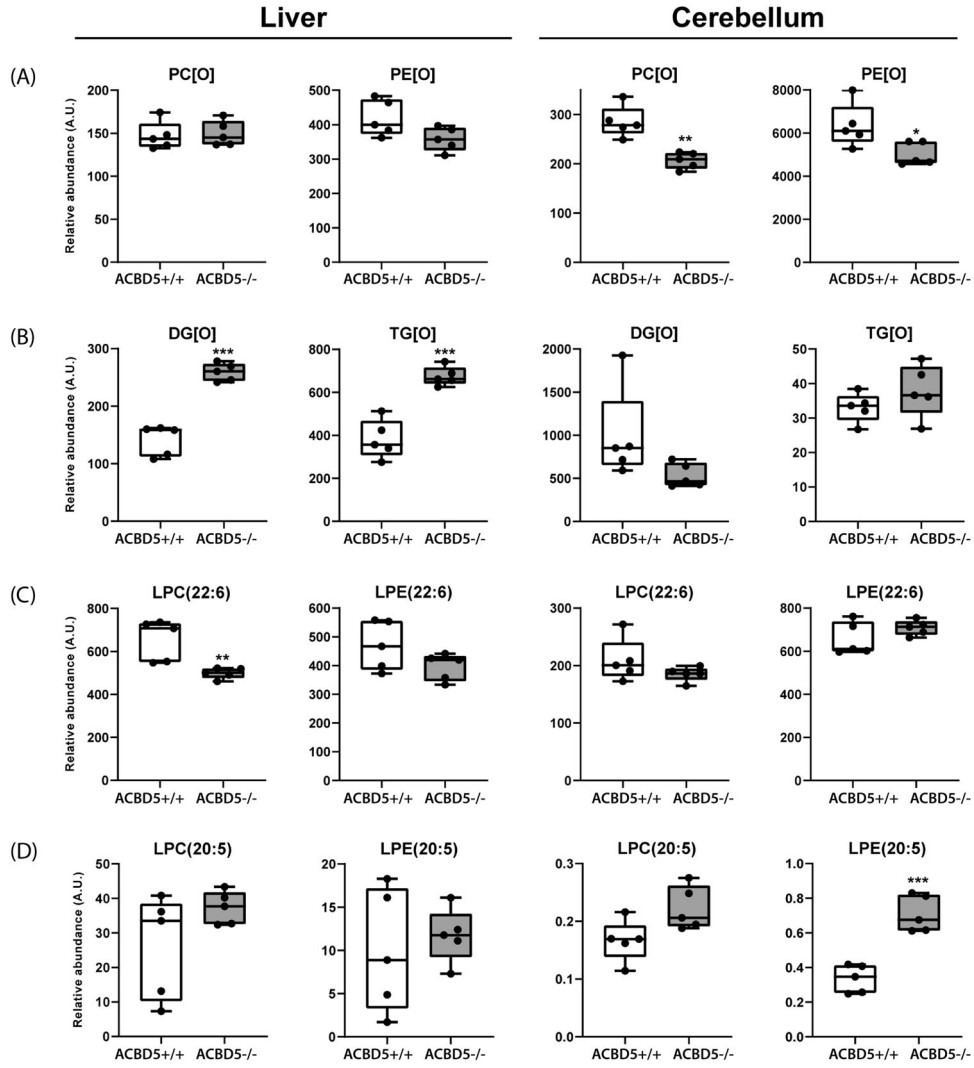

**Fig. 7 Alterations in major lipid classes in cerebelli and livers of *Acbd5*[−/−] mice.** Analysis of (**a**) ether phospholipids (PC[O], PE[O]) and (**b**) ether glycerolipids (DG[O], TG[O]) in livers and cerebelli of *Acbd5*[−/−] and *Acbd5*[+/+] male mice. **c** Quantification of docosahexaenoic acid-containing lyso-phosphatidylcholine (LPC(22:6)) and lyso-phosphatidylethanolamine (LPE(22:6)). **d** Quantification of eicosapentanoic acid-containg lyso-PC (LPC(20:5)) and Lyso-PE (LPE(20:5)). Total lipid levels are defined as the summation of the relative abundance of all identified phospholipid species of the same class normalized to the corresponding internal standard, assuming identical response with respect to internal standard. The whisker blots depict the median, interquartile range, and the maxima/minima (*$p < 0.05$, **$p < 0.01$, ***$p < 0.001$; unpaired *t*-test, two-sided).

Palmitic, oleic, and linoleic acids are the major FA in cellular lipids. Thus, an increase in PC with a total carbon chain length above C40 indicates elevations in PC with at least one VLCFA in its side chains (Fig. 8). Corroborating the findings from LPC, ULCFA containing multiple double bonds (mass>C60) were increased in cerebellar PC of *Acbd5*[−/−] mice. Remarkably, PC containing most likely two ULCFA (C72-C80) were almost exclusively detected in *Acbd5*[−/−] cerebelli. The results for PE and TG also show shifts towards longer, unsaturated FA in both organs (Fig. 8), but in contrast to PC, PE and TG profiles did not reveal major organ-specific differences and did not include the peculiarities in ULCFA. These findings indicate that ULCFA enriched in cerebelli of *Acbd5*[−/−] mice are specifically integrated into PC of cellular and subcellular membranes, but not into TG deposited in LD. Notably, oxidized ULCFA were also specifically enriched in PC of *Acbd5*[−/−] cerebelli (Supplementary Fig. 6B). Thus, the loss in ACBD5 appears to disrupt peroxisomal FA degradation, permitting FA elongation to non-degradable VLCFA and UCLFA at the ER.

## Discussion

Unlike other peroxisomal proteins mutated in single enzyme deficiencies, ACBD5 performs a second, non-metabolic function —the formation of membrane contacts with the ER[5,6]. Hence, the pathology of the ACBD5 deficiency could include a novel pathologic mechanism caused by the disruption of exchange processes between ER and peroxisomes. Moreover, the elevated VLCFA concentrations in ACBD5-deficient patients highlight its significance for peroxisomal β-oxidation[9,11]. Here we describe the phenotype of a mouse model for ACBD5 deficiency, and show by lipidomics that the loss of ACBD5 alters lipid parameters of additional peroxisomal pathways beyond FA β-oxidation. Our findings suggest that ACBD5, in addition to its role as an acyl-CoA binding protein, fulfills an important regulatory role at the crossroads of peroxisomal, ER and mitochondrial lipid metabolism by facilitating peroxisome-ER contact sites.

**The cerebellar phenotype of *Acbd5*[−/−] mice.** Judged by its progressive neuronal phenotype, the C57BL/6N-A[tm1Brd] *Acbd5*[tm1a(EUCOMM)Wtsi/WtsiCnbc] mouse (*Acbd5*[−/−]) has a similar

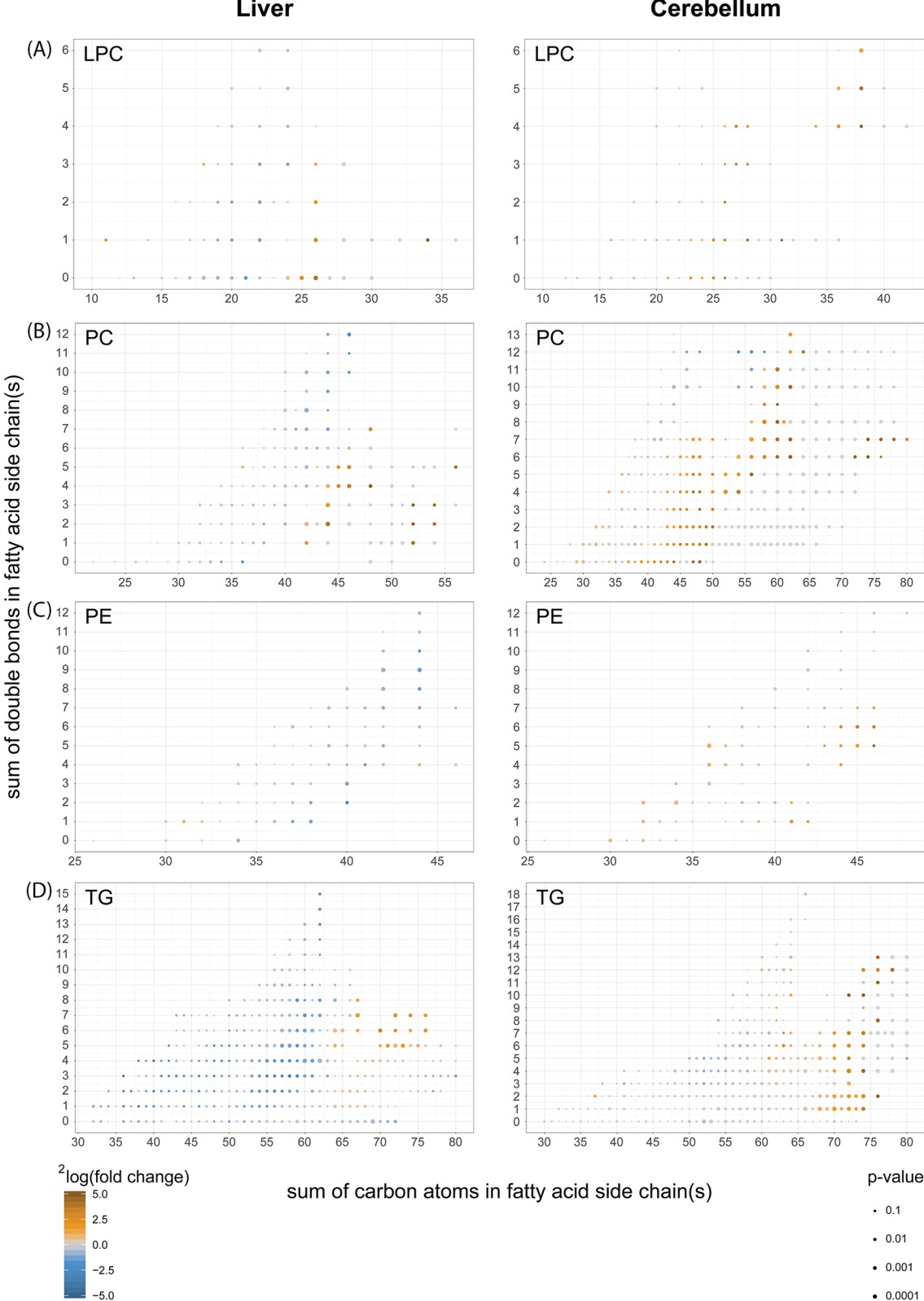

**Fig. 8 Fatty acid composition of phospholipids from cerebelli and livers of *Acbd5*⁻/⁻ mice.** Comparison of the fatty acid composition detected in (**a**) lyso-phosphatidylcholines (LPC), (**b**) phosphatidylcholines (PC), (**c**) phosphatidylethanolamines (PE) and (**d**) triacylglycerols (TG) from liver and cerebellum of *Acbd5*⁻/⁻ and *Acbd5*⁺/⁺ male mice (value obtained from 5 one-year-old males per experimental group); (*$p < 0.05$, **$p < 0.01$, ***$p < 0.001$, unpaired *t*-test, two-sided).

pathology as ACBD5-deficient patients. Indicating a cerebellar pathology, a demyelination in cerebellar peduncles has been described for ACBD5-patients[9]. Cerebellar atrophy and ataxia have been observed in several peroxisome disorders like some late-onset forms of ACOX1, D-bifunctional protein (DBP) and ABCD1 deficiency as well as in Refsum disease and RCDP[20]. Likewise, mouse strains with peroxisome deficiencies develop a progressive cerebellar neurodegeneration. These include knock-out mice for the β-oxidation enzymes DBP, ABCD2, and DHA-PAT involved in ether-lipid synthesis[20].

While $Dhapat^{-/-}$ mice show primarily a developmental phenotype, mouse models for peroxisomal β-oxidation deficiencies exhibit a degenerative cerebellar pathology. Thus, the progressive pathology in the $Acbd5^{-/-}$ model resembles $Dbp^{-/-}$, $Abcd1^{-/-}$ or $Abcd2^{-/-}$ mouse phenotypes. With respect to phenotype onset and pathologic alterations, $Dbp^{-/-}$ mice with a complete block in peroxisomal β-oxidation show a highly severe pathology[39,40] with massive cerebellar axon demyelination, astro- and microgliosis at an early age[39,41]. A brain-specific $Dbp^{-/-}$ mouse exhibits a similar, less severe pathology with a later onset[42,43]. In this respect, the moderate astro- and microgliosis, reduction in Purkinje cell number, altered AIS positions and swellings in Purkinje cell axons in $Acbd5^{-/-}$ mice point to a similar, less severe pathology implying that accumulating peroxisomal β-oxidation metabolites are a driving factor for phenotype development.

The ABC transporters ABCD1 and ABCD2 mediate peroxisomal acyl-CoA import with distinct but overlapping substrate specificities[44]. As ACBD5 might provide acyl-CoAs for peroxisomal ABCD transporters[11], a comparison of ABCD-deficient and ACBD5-deficient mice could give further information on the pathology in ACBD5 deficiency. $Abcd1^{-/-}$ and $Abcd2^{-/-}$ mice develop a mild, late-onset cerebellar phenotype with partial loss in Purkinje cells and reactive gliosis at advanced ages of 20 and 12 months, respectively[45,46]. An $Abcd1/Abcd2$ double-knockout strain develops a more severe locomotor phenotype with changes in cerebellar morphology observed at 12 months[45]. Accordingly, the phenotype of $Acbd5^{-/-}$ mice is more severe than in all three ABCD-deficient mice. However, $Abcd1^{-/-}$ and $Abcd1^{-/-}/2^{-/-}$ mice exhibit higher C26:0 FA concentrations than $Acbd5^{-/-}$ mice (5-6-fold vs. 2-3-fold elevation, Fig. 2b)[47,48]. Thus, VLCFA elevations do not correlate with phenotype severity. Therefore, additional peroxisomal pathways could be compromised in $Acbd5^{-/-}$ mice. Its progressive phenotype arising at 4–6 months suggests a continuous accumulation of hazardous lipid metabolites in the CNS inducing pathologic cellular alterations, which ultimately compromise the neurons' physiological functions. It is becoming increasingly obvious that membrane lipid composition can impact transmembrane protein structure and activity[49]. Hence, the altered membrane lipid composition in $Acbd5^{-/-}$ cerebelli might affect transmembrane protein structure in neuron-specific subdomains. In this context, subtle changes in the AIS position of Purkinje cells suggest molecular changes in the axonal compartment and might point to alterations in network activity. Further experiments are required to decipher if the AIS alterations are an effect of neuro-physiologic changes or directly correlated to an altered AIS architecture.

**$Acbd5^{-/-}$ mice exhibit a hepatic peroxisome proliferation untypical for a VLCFA β-oxidation deficiency.** While peroxisomal biogenesis disorders regularly induce prominent liver alterations, peroxisomal β-oxidation deficiencies show mostly minor hepatic changes[27]. Remarkable exceptions are the $Acox1^{-/-}$ and $Lbp^{-/-}/Dbp^{-/-}$ mouse strains. Both exhibit steatosis accompanied by the absence of functional peroxisomes at young age (5 weeks)[50,51]. While $Lbp^{-/-}/Dbp^{-/-}$ mice die

around this age, $Acox1^{-/-}$ mice develop a striking compensatory reaction over time, characterized by a massive PPARα-induced peroxisome proliferation in surviving hepatocytes[28,50]. Sustained activation of PPARα in $Acox1^{-/-}$ mice was further associated with liver tumor development[52]. $Acbd5^{-/-}$ mice do not develop a comparable hepatic pathology. Unlike in $Acox1^{-/-}$ and $Lbp^{-/-}/Dbp^{-/-}$ mice, peroxisome numbers were generally elevated in hepatocytes of young and old animals. Such an early hepatic peroxisome proliferation, was described for the $Scp2^{-/-}$ and $Amacr^{-/-}$ mouse lines, which both exhibit normal VLCFA concentrations, but elevated branched-chain FA[53,54]. Thus, the peroxisome proliferation observed in hepatocytes of $Acbd5^{-/-}$ mice does not appear to correlate with the elevation of straight-chain VLCFA but may be caused by an alternative metabolite.

**Tissue-specific lipidome changes support a function of ACBD5 in a metabolic hub between peroxisomes and the ER.** Our lipidome analysis revealed profound, tissue-specific alterations in lipid species. VLCFA content was elevated in almost all complex lipids. However, as reported for Zellweger and X-linked adreno-leukodystrophy (X-ALD) patients[55,56], ULCFA (≥C34), a minor FA component <1% in the brain of healthy individuals[57], were significantly increased in PC of $Acbd5^{-/-}$ cerebelli, but absent in liver. A comparable ULCFA increase was neither observed in PE, TG, phosphatidylinositols nor cholesteryl esters implying their selective integration into PC. Interestingly, elevated ULCFA in brains from Zellweger-spectrum disorder and X-ALD patients exhibited differing double bond numbers[55]. While polyenoic ULCFA was mainly elevated in PC from Zellweger patient brains, X-ALD patients rather contained increases in PC with monoenoic ULCFA[55]. A prevalent accumulation of monoenoic VLCFA in PC from $Abcd1^{-/-}$ mouse brains was likewise reported[58]. The lipid profiles from $Acbd5^{-/-}$ mouse cerebelli resemble therefore data from Zellweger, but not X-ALD patients. The more severe phenotype of $Acbd5^{-/-}$ compared to $Abcd1^{-/-}$ mice suggests that polyenoic ULCFA in PC might have a detrimental impact on cellular membrane properties. Moreover, the closer similarity between lipid profiles from $Acbd5^{-/-}$ mice and Zellweger patients implies that ACBD5 not merely channels VLCF acyl-CoAs for peroxisome import. Rather, the $Acbd5^{-/-}$ mouse phenotype might include the protein's tethering function. Disruption of peroxisomal β-oxidation not only impedes VLCFA degradation, but induces elongation of LCFA located at the ER membrane[59]. Knockdown of the FA elongase ELOVL1 in ABCD1-deficient fibroblasts reduced cellular C26:0 FA concentrations indicating that VLCFA levels are controlled by maintaining equilibrium between FA elongation and degradation[60]. In addition, substrate-specific FA elongases are differentially expressed in tissues. ELOVL4 elongating polyenoic FA > C28 is predominantly expressed in neuronal tissue, but not in liver[61]. Thus, unopposed ELOVL4 activities could explain the accumulation of ULCFA-containing PC in the cerebellum of $Acbd5^{-/-}$ mice and their absence in liver. FA composition of membrane lipids could thus be controlled via competitive degradation of VLCFA by peroxisomes opposing ER FA elongation[7,12]. In such a scenario, a spatial separation of ER and peroxisomes would promote VLCFA accumulation at ER membranes supporting their elongation to ULCFA (Fig. 9). Moreover, UCLFA are most preferentially incorporated into PC, but not other lipid classes. Consequently, ULCFA become incorporated into PC of cellular membranes eventually forming potentially hazardous lipid peroxidation products.

Ether phospholipids were decreased in cerebelli of $Acbd5^{-/-}$ mice. As the pathway of ether lipid synthesis is initiated

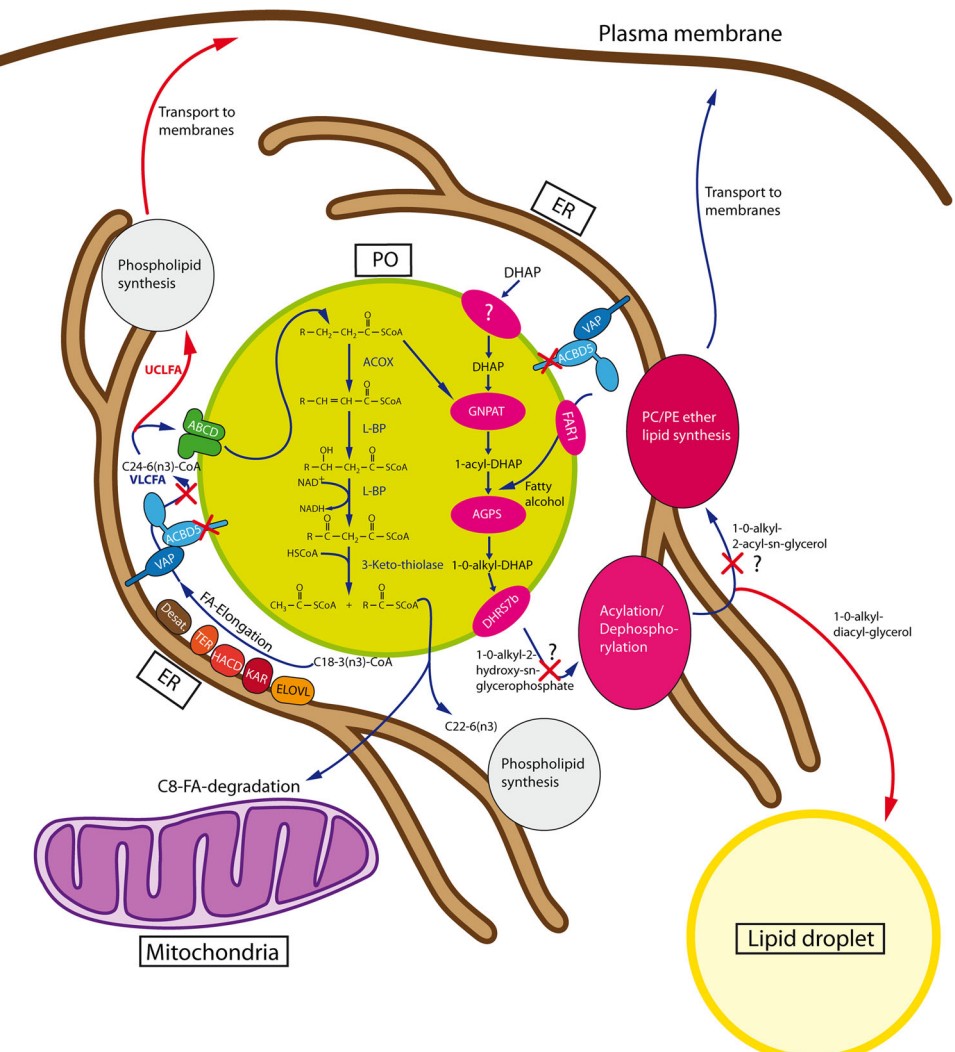

**Fig. 9 Alterations in lipid metabolism induced by the lack of ACBD5.** Fatty acid elongation is performed at the ER from stored LCFA. Peroxisomes might control FA over-elongation by a competitive import of VLCFA, requiring peroxisome-ER contact sites for their full productivity. ACBD5 disruption prevents peroxisomal import of VLCFA thereby accumulating substrates for elongation of VLCFA to UCLFA, which are subsequently incorporated into membrane lipids. Likewise the C24:6 n-3 precursor of DHA, which has to be chain-shortened by one round of peroxisomal β-oxidation is further elongated and desaturated to polyenoic PUFAs. Ether lipids are synthesized in peroxisomes up to the 1-O-alkyl-2-hydroxy-sn-glycerophosphate intermediates. Subsequent conversion to 1-O-alkyl-2-acyl-sn-glycerols and at last ligation of a PE or PC head group in sn3-position to yield the mature ether-phosphoglycerolipid is performed at the ER membrane. As revealed by lipidomics, ACBD5 disruption apparently induces formation of monoalkyl-diacyl-glycerols, which are ultimately rerouted to lipid droplets. Currently, the significance of peroxisome-ER contact sites is not fully understood but may contribute to a regulatory network controlling membrane ether-lipid formation.

in peroxisomes and completed at the ER[62], a reduction in peroxisome-ER contacts might impede the transfer of pathway intermediates. Alternatively, reductions in plasmalogen-rich myelin may account for the reduced ether lipids. Unexpectedly, $Acbd5^{-/-}$ mice contained highly increased amounts of ether lipid TG[O] and DG[O] in liver. According to current knowledge, peroxisomal ether lipid synthesis terminates with the synthesis of 1-O-alkyl-2-hydroxy-sn-glycerophosphate[62]. Subsequent acylation of the sn-2 and dephosphorylation of the sn-3 position are catalyzed at ER membranes forming DG[O][62]. Remarkably, instead of receiving its polar head group at the sn-3 position, the ether lipid intermediate DG[O] appear to be converted into TG[O] (Fig. 10). It is tempting to speculate if peroxisome-ER contacts are involved in regulating membrane ether lipid supply by a hitherto unknown mechanism. Alternatively, as the ACBD5 deficiency influences the fatty acid balance between peroxisomes and the ER, DG[O] FA composition could be altered, which

somehow changes the fate of DG[O] towards acylation to TG[O] instead of PE[O]/PC[O] synthesis.

According to the metabolic alterations, $Acbd5^{-/-}$ mice exhibit a complex phenotype potentially caused by a reduced peroxisomal β-oxidation and disrupted peroxisome-ER membrane contacts required for metabolite transfer and regulation. Future studies are required to explore the consequences of a reduction in peroxisome-ER contact sites in the pathology of ACBD5-deficient mice and humans as well as to delineate the mechanism inducing the degenerative alterations in the cerebellum and other CNS regions.

## Material and methods

**Animal breeding.** The C57BL/6N-A[tm1Brd] $Acbd5$[tm1a(EUCOMM)] [Wtsi/WtsiCnbc] mouse strain was recently produced at the Sanger Institute. In this mouse, ACBD5 transcription is blocked by inserting a neo cassette between exon 2 and exon 3 of the $Acbd5$

gene using the so-called knock-in first technology[63]. Sperm from $Acbd5^{+/-}$ mice was received from the European Mouse Mutant Archive (EMMA) and used for fertilization of wild-type C57BL/6 N mice ($Acbd5^{+/+}$). Heterozygous mice were identified by PCR (Table S1) and intercrossed to yield homozygous $Acbd5^{-/-}$ mice. As the $Acbd5^{-/-}$ mice are fertile, homozygous $Acbd5^{-/-}$ and correspondent $Acbd5^{+/+}$ mice received from the founder generation were further bred as homozygous strains (breeding & experiment permission under 35-9185.81/G-202/19, Regierungspräsidium Karlsruhe, Germany). On occasion, both strains were interbred and outcrossed to avoid genomic drift between both strains. The animals were kept according to the guidelines for care and use of laboratory animals of Germany at the animal facilities of the Medical Faculty Mannheim, Heidelberg University at a 12 h/12 h light cycle and were fed ad libitum with standard rodent chow. Animals were sacrificed by cervical dislocation for the extraction of tissue for biochemical analyses. For the preparation of material for histological and ultrastructural analyses, mice were sacrificed by intraperitoneal lethal dose of xylazine/ketamine (16/120 mg/kg b.w.).

**Animal testing**. The animals locomotory performance was examined using an established qualitative scoring system to rate symptoms for cerebellar ataxia[15]. The scoring system includes an evaluation of hind limb clasping, ledge test, gait, and kyphosis with each measure recorded on a scale of 0–3 thus yielding a combined total of 0–12 for all traits. Eight male and eight female $Acbd5^{-/-}$ and correspondent $Acbd5^{+/+}$ animals were scored at the age of 2, 4, 6, 9, 12, and 14 months in a blinded approach. At the same date, weight of each animal was determined.

**RT-qPCR**. Liver, testis, heart, kidney, cerebrum, cerebellum, and spleen from 6 $Acbd5^{-/-}$ as well as $Acbd5^{+/+}$ mice (age 2–3 months) was excised and directly frozen in liquid nitrogen ($n = 6$ per group). Approximately 20 mg of each tissue was homogenized in a bead mill (Beadrupter 4, Omni International, USA) using tubes with 2.4 mm metal beads. Total RNA was prepared using the RNeasy Mini Kit according to the manufacturer's instructions (Qiagen, Hilden, Germany). RNA integrity was validated by checking 28S and 18S RNA profiles using an RNA 6000 Nano Kit with a Bioanalyzer 2100 (Agilent, Waldbronn, Germany). For cDNA production, 1 μg of total RNA was reverse transcribed using the Biozym cDNA synthesis kit with oligo-dT primers according to the manufacturer's protocol (Biozym Scientific, Hessisch Oldendorf, Germany). Exon-spanning primers for ACBD5 were selected to avoid amplification from contaminating DNA (Supplementary Table 1). GAPDH was used as housekeeping gene product in order to calculate ΔCT values. The qPCR was performed with 0.5 μl of the cDNA reaction/run and Brilliant II SYBR® Green QPCR Master Mix (Agilent), 5 μM primer concentrations using a two-step cycle protocol with $40 \times 95\,°C/30$ s, $60\,°C/60$ s in a MX Pro 3005 P QPCR System (Agilent). After manual threshold setting ct (dr) (in log phase), the mean ct of the duplicates was calculated after correction via a housekeeping gene. Afterwards, the relative gene expression was normalized to liver tissue.

**Isolation of mouse embryonic fibroblasts**. Mouse embryonic fibroblasts (MEF) were isolated according to ref. [64]. Briefly, one pregnant $Acbd5^{-/-}$ and $Acbd5^{+/+}$ mouse with embryos (6 per litter) at stage E17 were sacrificed by cervical dislocation and the uteri excised and placed in a petri dish containing 100 U/L penicillin, 100 μg/L streptomycin in phosphate-buffered saline (PBS), pH 7.4. Subsequently, the embryos were removed from the uterus under a laminar flow hood. Head, heart and liver tissue of

all embryos was removed and the remaining tissue cut into pieces of 2–3 mm³, followed by digestion in 0.25%/0.02% trypsin-EDTA solution for 10 min at 37 °C. At this stage, cell suspensions from all respective $Acbd5^{+/+}$ and $Acbd5^{-/-}$ embryos were pooled. Subsequently, the cell suspension was pipetted up and down a 25 mL pipette to separate remaining tissue pieces into single cells and incubated for another 10 min. Thereafter, MEF culture medium (DMEM with 25 mM glucose, 10% fetal bovine serum, 100 U/L penicillin, 100 μg/L streptomycin) was added to the cell suspension for trypsin inhibition. Separation of remaining aggregates from mesenchymal cells was achieved by sedimentation of the cell suspension. After 5 min incubation at room temperature, the supernatant was seeded into cell culture flasks and cultured at 37 °C, 5% $CO_2$ in MEF culture medium. Subsequently, the cells were cultured for 5 passages and frozen as stock. To avoid, that differences in peroxisomes could be caused by ACBD5-independent mutations during the immortalization process, two aliquot stock solution were thawed and individually cultivated until the MEFs gained a homogenous fibroblast-like morphology and stable duplication times (>P 12) to be further used in the experiments. For the peroxisome elongation experiments, DHA was dissolved in 4% fatty acid-free bovine serum albumin in DMEM (Sigma-Aldrich) to supplement cell cultures at a final concentration of 150 μM. After incubation for 12 h and 24 h, respectively, cells were fixed for immunofluorescence microscopy (see below).

**Peroxisome isolation**. Peroxisome isolation from mouse liver tissue was performed as described previously[65]. In brief, for each isolation, liver from 6 male mice at the age of 2–3 months was dissected, pooled, and homogenized in ice-cold homogenization buffer (HB, 250 mM sucrose, 5 mM MOPS, 1 mM EDTA, 2 mM PMSF, 1MM DTT, 1 mM ε-aminocaproic acid, 0.1% ethanol, pH 7.4 adjusted with KOH) using a Potter-Elvehjem-type tissue grinder. The initial homogenate was centrifuged at $600 \times g_{av}$, 4 °C. The resulting pellet was rehomogenized and centrifuged using the same conditions. The supernatants from the $600 \times g$ centrifugation were pooled and centrifuged at $2700 \times g_{max}$, 15 min 4 °C to generate the heavy mitochondrial pellet. To increase yields, the heavy mitochondrial pellet was resuspended in HB and centrifuged with the same conditions. The supernatants from both runs were combined and centrifuged at $37{,}000 \times g$, 20 min. The reddish "fluffy" layer (FL) from the resulting light mitochondrial pellet was removed by hand and the remaining pellet washed once in HB using the same centrifugation conditions. The resulting light mitochondrial fraction enriched in peroxisomes was subsequently layered on a 1.12–1.26 g/mL Optiprep gradient of sigmoidal shape. The gradient was centrifuged in a vertical type VTi50 rotor (Beckman Coulter USA) at an integrated force of $1256 \times 10^6 \times g \times min$ with slow acceleration/deceleration). Peroxisomes are enriched in two bands at the gradient bottom at 1.18 and 1.23 g/mL. Individual bands were eluted with a syringe. The supernatant of the light mitochondrial fraction was further used for the production of a microsome-enriched fraction. To this end, the supernatant was spun at $100{,}000 \times g$ for 30 min, 4 °C. Protein content of all fractions was quantified by the Bradford assay, all fractions were kept frozen until further use.

**Immunofluorescence staining and microscopy**. All animals were perfused with 0.9% NaCl via the left ventricle to remove blood. For liver staining, the organs were removed, immersed in Tissue Tek® (Sakura Finetek) and immediately cryopreserved in liquid nitrogen. Cerebelli were perfusion-fixed with ice-cold 2% (synapse staining) or 4% PFA (cell staining). After overnight

immersion in 2% or 4% PFA at 4 °C, respectively, cerebelli were cryoprotected by sequential incubation in 5% (2 h), 10% (2 h) and 30% (overnight) sucrose and subsequently embedded in Tissue Tek® (Sakura Finetek). For the detection of cells, immunofluorescence was performed on free-floating 50 μm thick cryostat sections. 20 μm sections, used for the detection of subcellular structures, were stained directly on slides. A detailed description of all primary and secondary antibodies used in this study is given in Table S2. For immunostaining, all primary antibodies were diluted in blocking buffer (1% BSA, 0.2% fish skin gelatin, 0.1% Triton X-100 in TBS) and incubated at room temperature overnight. Secondary dye-labeled Alexa antibodies (Fluor 488, 568 and 647-labeled IgG, Invitrogen) were incubated in blocking buffer for 1.5 h using antibody dilutions specified in Table S2. To-Pro-3-iodide (TOPRO, Invitrogen) was used as nuclear stain at 1:1000. Omitting the primary antibody completely abolished all stainings.

Confocal analysis was performed on a C2 Nikon confocal microscope equipped with 488 nm, 561 nm and 647 nm laser lines and either an ApoPlan 20× (0.75 NA), ApoPlan 60x (oil immersion, 1.4 NA) or a ApoPlan 100× (oil immersion, 1.45 NA) objective. Thickness of single optical sections was 1 μm for the imaging of whole cells and 0.5 μm for subcellular structures (synapses, peroxisomes) in stacks of 10–20 μm total depth. For the quantification of cell numbers, stacks of images were merged into a maximum intensity projection. For the quantification of synapse and peroxisome densities, single planes were used. Counting, size, relative area, and circularity determination of fluorescent signals in MEF and tissue sections were performed with Fijii ImageJ using automated thresholding and the "analyze particles" command (size: 2 pixels-infinity, circularity: 0.0–1.0)[66]; for contrast enhancement, image tif-files were processed in Adobe Photoshop CC (Adobe Systems). AIS measurements were performed as previously described using a self-written GUI application utilizing Python and the open-source modules matplotlib, opencv, python-bioformats, pyqt and numpy[67]. Briefly, axons were traced through a graphical interface starting from the soma, passing through the AIS and ending where the color trace of the AIS channel was no longer perceivable to the human eye. The image was then rectified along this axis through affine transformations. Length was evaluated by calculating the pixel distance of start and end, establishing the proximal and distal boundaries of the AIS as the first and respectively the last pixel value over 0.3 on the normalized intensity curve of the AIS channel. Distance to soma was provided by the distance from the start of the traced line to the proximal AIS signal.

**Histological staining**. Liver tissue was prepared as described above (5 animals/group). For the evaluation of gross liver anatomy, 10 μm thick cryostat sections were post-fixed with 4% PFA for 10 min and stained with hematoxylin/eosin using standard protocols. Sudan Red IV staining was performed according to Herxheimer's technique. In brief, cryosections were postfixed with 4% PFA, 10 min. After washing, tissue sections were stepwise incubated in 50% ethanol, 2 min, Sudan Red IV solution, 2 min and shortly rinsed with 70% ethanol. Hematoxylin was used as a counterstain of nuclei. Alternatively, sections were stained with hematoxylin and eosin. All tissue sections were mounted using Rotimount Aqua (Carl Roth GmbH, Karlsruhe, Germany). Images were acquired using a Zeiss Ax10 microscope equipped with EC Plan-Neofluar objective (×40 magnification, NA 0.75) and an AxioCamHRC (Carl Zeiss Microscopy GmbH, Jena, Germany). Lipid droplets were quantified for an area of 0.075 mm²/animal.

**Electron microscopy and 3D reconstruction from serial sections**. Livers used for ultrastructural studies were perfused via the portal vein using 0.9% NaCl to remove blood ($n = 3$ animals). Perfusion was thereafter immediately switched to 1.25% glutaraldehyde in 0.1 M cacodylate buffer, pH 7.2 containing 0.05% $CaCl_2$ for 10 min. Livers were excised, cut into smaller pieces, rinsed in 0.1 M cacodylate and sliced into 100 μM thick sections using a Vibratome (Mikrom HM 650 V, Thermo Scientific). Subsequent DAB staining was performed as described[68] with the exception that the DAB incubation with $H_2O_2$ was extended to 2.5 h. Membranes in tissue slices were stained en bloc as described previously[69]. In brief, samples were incubated for 12–15 h in cacodylate buffer (0.1 M, pH 7.4) followed by 1 h in ferrocyanide (1.5%) and osmium tetroxide (2%). Afterwards, samples were incubated sequentially in thiocarbohydrazide (saturated solution, 20 min), osmium tetroxide (1%, 30 min), and lead aspartate (20 mM, 30 min). The samples were rinsed with water after each incubation step. All incubations were performed at room temperature except lead aspartate, which was used at 60 °C. After the staining, samples were dehydrated in solutions of ascending alcohol concentrations followed by incubation in propylene oxide. Epoxy resin was used for embedding. The resin was polymerized at 60 °C for at least 36 h. Serial sectioning was performed as described before[33]. Ultrathin serial sections (40–60 nm) were cut on an Ultracut S ultramicrotome (Leica) using a diamond knife (Diatome). The serial sections were collected on silicon wafers (Si-Mat SiliconMaterials) and sectioning induced compression was neutralized by exposure to chloroform vapor. Scanning electron microscopy was performed with a LEO Gemini 1530 equipped with a field emission gun and an ATLAS scanning generator (Zeiss). Images of 8500 × 8500 pixels (3.5 nm resolution) were taken of the same area in consecutive sections using the SE2 detector. Imaging parameters were as follows: 4.7 mm working distance, 30 μm aperture and 2 kV acceleration voltage. For the quantification of peroxisomes and ER contacts, 0.01 mm² of each liver were evaluated. Organelle sizes and distances were measured with Fiji[66]. Organelle abundance and contact site numbers were manually counted using image stacks blinded for the observer. Serial scanning electron microscopy images of the same field of view were used for 3D reconstruction using OpenCAR software[70]. Micrographs of 15 consecutive sections (900 μm²/section) were aligned and the structures of interest were manually traced in each section. Traced structures were reconstructed using the Delaunay-method (see also Satzler et al. 2002). The obtained surface renderings were analyzed for surface, volume and the closest distance between objects in 3D using custom written Matlab routines. Statistical significance was assigned by Student's $t$ test using Prism 5.0 software (GraphPad). Data are presented as mean ± SEM.

**Measurement of plasma and tissue fatty acid concentrations**. For the determination of lipid metabolites 5 $Acbd5^{-/-}$ and $Acbd5^{+/+}$ per gender were sacrificed. VLCFA were determined according to a modified protocol of[71] by gas chromatography-mass spectrometry (GC-MS). In brief, the tissue samples were transferred into 10 ml-glass tubes, suspended in 200–1000 μL of aqua dest., and then homogenized with a rotor stator homogenizer (IKA Ultra-Turrax, Staufen, Germany). 100 μL (plasma and homogenates) were methylated with 1 mL of methanolic HCl (3 M) at 80 °C. After 1 h, the reaction was stopped by cooling down to room temperature. The mixture was extracted two times with 2 ml of hexane by shaking for 20 min. The combined organic phases were evaporated to dryness at 60 °C with a stream of nitrogen and then reconstituted in 50 μl dichlormethane. For GC-MS analysis, the quadrupole mass spectrometer MSD 5972 A (Agilent, Santa Rosa, California, USA) was

run in the selective ion-monitoring mode. Gas chromatography separation was achieved on a capillary column (DB-5MS, 30 m × 0.25 mm; film thickness: 0.25; J&W Scientific, Folsom, California, USA) using helium as a carrier gas. A volume of 1 µl of the derivatised sample was injected in splitless mode.

Acylcarnitines were determined in plasma and homogenates by electrospray ionization tandem mass spectrometry (ESI-MS/MS) according to a modified method as previously described (Sauer et al. 2006) using a Quattro Ultima triple quadrupole mass spectrometer (Micromass, Manchester, UK) equipped with an electrospray ion source and a Micromass MassLynx data system. In brief, 5 µl plasma/tissue homogenates were given on a 4.7 mm filter paper punch, dried at room temperature overnight and extracted with 100 µL of deuterium labeled standard solution in methanol. After 20 min, the samples were centrifuged and the extract was evaporated to dryness, reconstituted in 60 µL of 3 N HCl/butanol, placed in sealed microtiter plates, and incubated at 65 °C for 15 min. The resulting mixtures were dried, and each residue was finally reconstituted in 100 µl solvent of methanol/ water/ (50:50 v/v) prior measurement. The resulting total amounts of the acylcarnitines of the homogenates were normalized to total protein.

Plasmalogens were determined as their dimethylacetals by gas chromatography flame ionization detection (GC-FID) according to a modified previously published protocol[72]. 100 µL of washed erythrocytes were mixed with 100 µL distilled water and 4–5 glass beads and shaken for 1 min. After 10 min incubation with 3 mL 2-propanol, 2 mL chloroform was added to the samples. After 10 min, the samples were centrifuged for 5 min at 1500 g and the upper phase containing the plasmalogens was transferred in a 10 mL-reaction vessel. Plasmalogens were methylated with 2 mL of methanolic HCl (3 M) at 80 °C. After 1 h, the reaction was stopped by cooling down to room temperature. 2 mL of potassium carbonate solution (14% w/v) and 2 mL of hexane was added and the mixture was shaken for 15 min. After 5 min centrifugation at 1500 g, 300 µL of the hexane phase was transferred into a vial, dried down and reconstituted into 60 µL hexane. Gas chromatography separation was achieved on a GC-FID (Carlo Erba Instruments HRGC 5300 Mega series) with a capillary column (Agilent CP-Sil 88 for FAME, 50 m/0.25 mm/ 0.2 µm, # CP 7488) and helium as a carrier gas.

Urinary bile acid metabolites were determined as previously described[73]. In brief, to a fixed 300 µl urine volume 20 µl internal standard solution (100 µmol/L D4-glycocholic acid) was added. For the solid-phase extraction the Oasis™ HLB (Hydrophilic-Lipophilic Balance) cartridges from Waters were used. The cartridges were conditioned with 2 × 1 mL dichlormethane/ methanol (2:1, v/v) and equilibrated with 1 mL purified water. Then the urine sample was loaded onto the cartridges and washed with 2 × 1 mL purified water and 2 × 1 mL n-hexane. The cartridges were kept at room temperature for 30 min to let the remaining n-hexane evaporate and bile acid conjugates were eluted with 300 µL of 700 mL/L aqueous methanol (v/v). The eluate was 10X diluted with aqueous methanol (700 mL/L, v/v) prior to injection. The HPLC system was operated isocraticallly at 80 µL/min flow rate for the mobile phase (acetonitrile/purified water, 1:1) at room temperature. Liquid nitrogen was used as the desolvation and nebulizer gas and argon was used as collision gas. Negative ion mass spectra of the eluents were recorded. Sample was injected in the LC-MS/MS by automatic injector.

**Western blotting**. Equal protein sample amounts (5–10 µg for tissue fraction, 20 µg for homogenates) were separated on 12% SDS gels and transferred to PVDF membranes using a semidry discontinuous blotting system. After transfer the membranes were blocked in 5% fat-free milk powder in PBST. Primary antibodies (Supplementary Table 2) were incubated overnight at 4 °C. After washing secondary HRP-conjugated antibodies were incubated for 1.5 h at room temperature. Chemiluminescent signals were produced with ECL and monitored with a Fusion Solo S Western blot imaging system (Vilber-Lourmat, Marne-la-Vallée, France).

**Tissue lipidomics**. For lipidomics experiments, liver and cerebellum of 1-year-old male mice (5 $Acbd5^{-/-}$ and 5 $Acbd5^{+/+}$ controls) were excised and directly frozen in liquid nitrogen. Tissue homogenates were made in Milli-Q water in a Qiagen Tissuelyser II using stainless steel beads of 5 mm for two times 30 s at 30 revolutions per second. The protein concentration of the homogenates was determined with the BCA assay. Lipid extraction and subsequent lipidomics analysis and data processing were performed essentially as described by ref. [36] by the Core Facility Metabolomics of the Amsterdam UMC.

**Reporting summary**. Further information on research design is available in the Nature Research Reporting Summary linked to this article.

## Data availability
All relevant datasets obtained and analyzed in the current study can be found in the Supplementary Data 1 or are available from the corresponding author upon request.

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

 ARTICLE

## Acknowledgements

The authors thank D. Türker, A. Uhl, S. Vorwald, H. Mohr, M. Vervaart and F. Kratzer for technical assistance. We further acknowledge the support of Dr. M. Engelhardt in AIS quantitation. We much appreciate the scientific discussions and manuscript corrections of R. Wanders, M. Schrader, J. Costello and A. Völkl. This work was supported by the Deutsche Forschungsgemeinschaft DFG, grant No. 397476530 (to M.I.) and the MEA-MEDMA Anschubförderung of the University of Heidelberg (to M.I.). This work was supported by the LIMA Core Facility Unit of the CBTM, University of Heidelberg.

## Author contributions

Conceptualization: M.I., F.M.V., H.R.W.; Methodology: C.K., H.H., T.K., F.M.V., M.I., J.G.O., K.V.S.; Software: J.R., H.Z., A.H.C.K; Investigation: W.D., M.S., J.L., Ö.S., S.K., G.D., K.V.S., H.H., S.H., M.I., F.M.V., Data curation: M.L.P., A.H.C.K.; Data analysis: W.D., M.S., J.L., G.D., C.K., M.L.P., F.M.V., M.I., K.V.S., Resources: H.R.W., F.M.V., T.K., J.G.O., C.S., M.I.; Writing—original draft: M.I., F.M.V., K.V.S., C.K.; Writing—Review & editing: M.I., C.S., F.M.V., C.K., H.R.W., Visualization: W.D., M.S., J.L., Ö.S.; F.M.V., M.L.P., M.I.; Supervision: M.I., F.M.V., J.G.O.; Project administration: M.I.; Funding acquisition: F.M.V., M.I.

## Funding

## Competing interests

The authors declare no competing interests.
