## [Peer Review File · Communications Biology]

Reviewers' comments:

Reviewer #1 (Remarks to the Author):

In this manuscript, an extensive description of the phenotype of *Acbd5* knockout mice is reported ranging from peroxisomal appearance in MEFs, pathology in cerebellum to lipidome analysis. ACBD5 is a peroxisomal membrane protein, which was shown to be involved in tethering peroxisomes to the ER, but was also suggested to facilitate transfer of fatty acids into peroxisomes via its acyl-CoA binding properties. Several data are in line with the findings in patients (ataxia, increased levels of VLCFA, reduced number of ER-peroxisome contact sites) whereas other data are novel (increased levels of ULCFA in cerebellum). The data are well illustrated and convincing, the lipidomics is impressive. In the discussion, an extensive comparison with other peroxisomal mouse models was made, trying to link metabolic deregulation with pathology, which is interesting. This paper is certainly an important addition to the knowledge on peroxisome biology. However, there is a lack of some data and some conclusions are not supported by the data.

Overall, the manuscript appears a bit heterogenous. Experiments are conducted in MEF, cerebellum and liver but several parameters were not looked at in all cell types. For example, peroxisome appearance was investigated in MEF and liver, but not discussed at all in cerebellum and peroxisome shape only in MEF. Histology was done in two different organs but nothing was mentioned on the integrity of other tissues e.g. are the mice fertile? Although the role as tether was further supported, its involvement as a protein facilitating import of peroxisomal β -oxidation substrates was not substantiated. The titles and content of paragraphs is not always coherent.

Major comments

The title does not reflect well the contents of the paper. "Neurological phenotype" gives the impression that the nervous system was thoroughly analysed, which is not the case. It is recommended to be more specific.

The statement in the abstract "ether phospholipid synthesis was perturbed in liver" is not supported by the data in Fig 8

The PEX14 staining in Fig 1C leads the authors to conclude that peroxisome biogenesis functions normally in *Abcd5*^{-/-} MEFs. This is not correct because disrupted peroxisomal matrix import will still allow PEX14 incorporation in the peroxisomal membrane.

Were the experiments with MEFs conducted on cells derived from one single wild type and knockout mouse? After 16 passages of primary mouse fibroblasts, the cells are in fact immortalized and serial passaging effects could occur. Cells should be used at low passage numbers and/or independent biological repeats should be done.

Paragraph on cerebellar phenotype : From the lipid analysis, it is concluded that there is a typical β -oxidation disorder, but the levels of DHA, that might have an effect on peroxisome size and shape, were not quantified. Line 125: it is stated that phytanic acid was measured but data were not shown, nor discussed. More importantly, bile acid intermediates are particularly important substrates for peroxisomal β -oxidation in liver, and nothing is mentioned on bile acid species and levels in the manuscript. This is essential to obtain a comprehensive view on the lipid changes. The cerebellar phenotype was investigated in great detail but no information is given on the other brain areas, leaving the reader in doubt whether there is no pathology or whether this was not looked at. This information should definitely be included. This is also prompted by the observation that the loss of Purkinje cells seems to be mild, whereas the kyphosis seems to be pronounced, raising the question whether other pathologies occur in the nervous system.

The title of the paragraph starting on line 198 is not well chosen. According to the conclusion it is intended to support the function of ACBD5 as an acylCoA binding protein delivering substrates to the peroxisome. The reasoning from line 241- 245 is not clear at all. It is impossible to deduce from normal ACOX1 activity that peroxisomal β -oxidation functions normally. In addition, how can it be concluded from these data that ACBD5 supplies Acyl-CoAs to the transporters? Overall, the message and the data that ACBD5 functions as a protein providing acylCoA to peroxisomal β -oxidation is not supported and conveyed well. It is quite unusual that the number of peroxisomes increases (proliferation) while the peroxisomal enzyme expression levels are unchanged. Are the authors aware of any other published condition with this discrepancy between peroxisome number

and enzyme expression levels?

There are several changes with age in the mutant mice: body weight, lipid droplets in liver. Can this not be related to the aggravating motor disability? Was the food intake monitored? This should be incorporated in the text. Line 215: the conclusion 'LCFA preferentially incorporated into TG for lipid storage' is unclear.

The analysis of cerebellar degeneration in combination with lipidomics is very interesting. Do the authors have any idea on the absolute levels of the VLCFA in cerebellum?

The impact of ACBD5 deficiency on ether lipid content might have to be interpreted more carefully. There was no shortage of plasmalogens in erythrocytes, and no deficiency of PC[O] and PE[O] in liver. Can the reduction of the latter species in cerebellum not have to do with reduced levels of myelin in this tissue at the age of 1 year? Myelin is indeed the most important lipid fraction in the brain. This should be reconsidered throughout the manuscript. To draw firm conclusions, one should know whether the metabolic defects precede or are a consequence of the pathology. Importantly, in my version of Fig 8, I do not find the statistical analysis that is mentioned in the figure legend. This prevents the assessment of these data for example elevation of TG[O]. It would be informative to also analyse TAG and DAG and see how they compare with DG[O] and TG[O] levels in the same tissues. Because the Y-axis of each panel of Figure 8 starts differently, attention should be drawn on this in the figure legend. Overall the reduction in LPC-22:6 is very modest, and it can hardly be claimed that its synthesis strongly depends on contact sites between ER and peroxisomes. What is the total DHA content in the tissues and in plasma?

Minor comments

The authors should comply with the guidelines for gene and protein nomenclature in men and mice. Although the methods were very extensively described, I do not find any explanation or reference how the quantification of peroxisome number, size and shape in MEFs was performed.

In addition to the MBP western blot, it would be informative to show an overview staining of the white matter in the cerebellum using MBP antibodies.

Line 141: and important role

Line 324: "only in the cerebellum....exhibited"

Fig7A ACBD3 should be ABCD3

Presentation of the data in the results section is not always chronologically followed in the figures, in particular figures 5,6 and 7 on liver phenotype and for example lipidomics data. It would be more logic to comment on DHA/DPA prior to homeostasis of VLCFA.

In the discussion on peroxisome proliferation in liver, it is recommended to elaborate on the discrepancy between proliferation and unaltered levels of peroxisomal enzymes.

Reviewer #2 (Remarks to the Author):

The manuscript entitled "The neurological phenotype 1 in ACBD5-deficient mice is associated with unexpected alterations in cellular lipid homeostasis" by Darwisch and colleagues is of great importance for our understanding of organelle interaction and even more importantly to understand what happens if this interaction is disturbed. Very convincing data showing robust effects in peroxisomal alterations, biochemical defects, and cerebellar abnormality's accompanied by a clear phenotype. The techniques used are state of the art and the manuscript is very well written.

There are only minor comments:

1) It is a pity that there is no statistical information provided in Fig 8. It would be interesting whether or not PE-ether are equally significantly reduced in liver and cerebellum – just from the graph this might well be! For the PC-ethers it is obvious without statistics. In general I would suggest to use statistical analysis to prove that the change between wt and ko is different in those lipid classes that are finally used for the argumentation of the tissue specific different lipid alterations.

- 2) In Fig 3A and B it should be indicated in which colour calbindin, GFAP and MBP is stained. In particular as the figure legend indicate that the arrow in B highlight axonal swellings of Purkinje cells (one in blue and one in red)?
- 3) In line 213 it is referred to Fig. 8 to see the overall decreased of liver TG levels but there are only the ether TGs and these are increased.
- 4) In the Material section it should be indicated in which solvent and how DHA was added.
- 5) It is unclear of how many different embryonic cell lines from ko and wt have been used for the imaging. It would be appreciated if the method used for quantification in Fig. 1E to H could be described in more detail.

Reviewer #3 (Remarks to the Author):

The manuscript by Darwisch et al presents the characterization of a novel mutant mouse line - Acyl-CoA binding domain containing protein 5 (ACBD5) null mice - a model of ACBD5 deficiency, a rare human disorder. The pathology associated with the condition includes, but is not limited to, retina dystrophy and motor problems that are manifest in early childhood. Loss of ACBD5 expression in patients was shown to result in increased abundance of very long-chain fatty acids (VLCFAs) linked to peroxisomal β -oxidation deficits. Despite these advances the pathogenic mechanisms of ACBD5 deficiency remain unclear. Here the authors provide a comprehensive characterization of ACBD5^{-/-} mice by examining motor performance, lipidomic profile and peroxisome properties at the cellular, biochemical and ultrastructural level. Through these analyses the author show that some phenotypic abnormalities reported in patients are recapitulated in the mouse model and report additional metabolic and cellular alterations linked to the mutation.

The work is of interest in that it characterizes a new mouse strain that could be of value as disease model and beyond that for the understanding of peroxisome biology. The fact that the mutant mouse recapitulates some of the pathologies in patients gives confidence in its value as disease model. The study is fairly comprehensive, with phenotypic analyses at multiple levels. Behavioral tests to assess motor deficits are adequate to conclude that a prominent pathology linked to the condition (motor deficit, retinal dystrophy was not tested) is recapitulated in the mouse model albeit with much delayed manifestation compared to patients. Analysis of cerebellar pathology provides some evidence of a mild neurodegenerative process with Purkinje cell loss. Lipidomic studies are thorough and of value in that they identify novel tissue-specific deficits by comparing liver tissue vs. cerebellum. Furthermore analysis of peroxisome biogenesis in the liver, including at ultrastructural level, convincingly show overabundance of peroxisomes in absence of gross tissue abnormalities, an intriguing novel finding.

SPECIFIC COMMENTS

The analysis of peroxisome-ER contact sites was carried in hepatocytes but not in other cells (MEF or cerebellum). It would be of high interest to examine contact sites in the cerebellum given evident pathology and differences observed with lipidomics. These experiments would also establish whether presumptive lack of contact sites between peroxisome-ER is a general feature of all cells lacking ACBD5.

A particular emphasis throughout the manuscript is placed on the potential role of deficits in peroxisome-ER membrane contact sites, induced by absence of ACBD5, in cellular and metabolic perturbations. However experiments to establish causal relationship are not presented. Because of this, although an attractive hypothesis, such emphasis seems unwarranted.

An attempt is made to examine myelination by measuring expression of PLP and MBP (Fig. 6) but these data are not appropriately quantified and therefore inconclusive.

The protein composition of peroxisomes in liver was examined biochemically (Fig. 7A-C) but no

statistical analysis is reported to buttress these observations (eg ACBD3, ACOX1, VAPB).

Expression of ACBD4 protein level, which could play a compensatory role, was not examined (only mRNA was measured).

We would like to thank the referees for their positive reviews of our manuscript entitled "The neurological phenotype in ACBD5-deficient mice is associated with unexpected alterations in cellular lipid homeostasis".

Please find a revised version of our manuscript enclosed. We have incorporated the suggestions of the referees as indicated below. The revised manuscript contains five novel supplementary figures (Fig. S1 – Fig. S5), specifying peroxisome status in MEFs, hepatocytes and cerebellar Purkinje cells and illustrating the architecture of the retina as well as graphs for food consumption rates and additional lipid metabolic data. We further added four figures (Fig. R1 – R5) as further proofs for the referees not intended for publication.

We hope that our manuscript can now be accepted for publication in "Communications Biology".

Response to REVIEWER(S)' COMMENTS

Reviewer #1 (Remarks to the Author):

In this manuscript, an extensive description of the phenotype of *Acbd5* knockout mice is reported ranging from peroxisomal appearance in MEFs, pathology in cerebellum to lipidome analysis. ACBD5 is a peroxisomal membrane protein, which was shown to be involved in tethering peroxisomes to the ER, but was also suggested to facilitate transfer of fatty acids into peroxisomes via its acyl-CoA binding properties. Several data are in line with the findings in patients (ataxia, increased levels of VLCFA, reduced number of ER-peroxisome contact sites) whereas other data are novel (increased levels of ULCFA in cerebellum). The data are well illustrated and convincing, the lipidomics is impressive. In the discussion, an extensive comparison with other peroxisomal mouse models was made, trying to link metabolic deregulation with pathology, which is interesting. This paper is certainly an important addition to the knowledge on peroxisome biology. However, there is a lack of some data and some conclusions are not supported by the data.

Overall, the manuscript appears a bit heterogenous. Experiments are conducted in MEF, cerebellum and liver but several parameters were not looked at in all cell types. For example, peroxisome appearance was investigated in MEF and liver, but not discussed at all in cerebellum and peroxisome shape only in MEF.

Unfortunately, word count restrictions do not allow to discuss findings from all tissues in detail. Therefore, we focused the manuscript text on findings, which were most evident in a particular cell type. To complement the data shown, we added novel graphs for peroxisomes in liver showing peroxisome size distribution as well as peroxisome length/width ratios based on the EM data in order to describe potential changes in organelle appearance in the revised version of the manuscript (Fig. S4 E+F). In line with the findings from MEF, peroxisomes in *Acbd5*^{-/-} hepatocytes show shift towards smaller diameters and a reduced length/width ratio implying that less peroxisomes occur in an elongated form. A thorough description of peroxisome appearance in the cerebellum is more complicated. Peroxisome size and shape can vary substantially in different cell types. Therefore a detailed description on morphological changes of peroxisomes in the cerebellum would require their separate analysis in numerous cell types. Moreover, in neurons, organelles in neurites can have a different size or shape as in the soma. Therefore, we focused in a novel supplemental figure on Purkinje cell somata to describe the peroxisome status in the cerebellum by immunofluorescence staining (Fig. S3). Quantifications of peroxisome numbers and percentage area in Purkinje cells are

now given. Unlike in hepatocytes or MEF, peroxisomes, like most other organelles in neurons, accumulate in the narrow perinuclear region of the soma, thus precluding a detailed analysis of peroxisome morphology by confocal microscopy (see Fig. S3). We attempted to analyze peroxisome morphology by EM applying alkaline DAB staining, however, the low catalase content of peroxisome from Purkinje cells (see immunofluorescence images) prevented a reliable identification of peroxisomes. We did as well not succeed with immunogold labeling via PEX14 antibodies applying the Tokayasu method. While we able to label vesicles with a peroxisome like morphology, poor membrane preservation did not allow a detailed analysis of peroxisomal shape and frequency of membrane contact sites (see Fig. R3+R4)

Histology was done in two different organs but nothing was mentioned on the integrity of other tissues e.g. are the mice fertile?

For this manuscript, we selected for our analysis two tissues, liver, as an organ with highest peroxisome abundance/metabolism and cerebellum according to the behavioral phenotype of the *Acbd5*^{-/-} mice. Until now, we did not observe further gross anatomical changes in other organs but as Communications Biology's publication policy precludes the description of observations with "data not shown or to be published independently", we avoided to include such general statements on potential pathologic alterations into the manuscript. As described in the "Methods" section, homozygous *Acbd5*^{-/-} mice were used for breeding, which presumes that the animals are fertile. Nevertheless, we agree with the reviewer, that such a detail can be easily missed by the reader and included a statement on fertility into the results section (P. 5, L. 124). Since patients with ACBD5-deficiency develop a striking retinal dystrophy, we considered that a small section on this tissue should be added to an initial publication describing a corresponding mouse model. Therefore, we added a small supplemental figure (Fig. S4), which show, that the retina is as well target of pathologic alterations in the *Acbd5*^{-/-} mouse. In the future, more details on the retinal pathology will be published in an independent paper.

Although the role as tether was further supported, its involvement as a protein facilitating import of peroxisomal β -oxidation substrates was not substantiated.

Please see response to the comments on β -oxidation below (answer to reviewer 3).

The titles and content of paragraphs is not always coherent.

Some titles were modified, as indicated in the revised manuscript, to better summarize the content or their paragraphs.

Major comments

The title does not reflect well the contents of the paper. "Neurological phenotype" gives the impression that the nervous system was thoroughly analysed, which is not the case. It is recommended to be more specific.

We specified the title accordingly to "The cerebellar phenotype in ACBD5-deficient mice is associated with unexpected alterations in cellular lipid homeostasis".

The statement in the abstract "ether phospholipid synthesis was perturbed in liver" is not supported by the data in Fig 8

We apologize for this mistake. Indeed, in liver ether lipid synthesis is perturbed with respect to the increased amount of ether glycerolipids. We corrected the abstract accordingly to "ether lipid synthesis"

The PEX14 staining in Fig 1C leads the authors to conclude that peroxisome biogenesis

functions normally in *Abcd5*^{-/-} MEFs. This is not correct because disrupted peroxisomal matrix import will still allow PEX14 incorporation in the peroxisomal membrane. Immunofluorescence signals for catalase, ACOX1, Pex3 highly colocalize with Pex14 in peroxisomes in *Abcd5*^{-/-} and *Abcd5*^{+/+} MEFs. We did not observe an obvious change in this peroxisomal protein distribution upon the DHA treatment, when IF signals of catalase and Pex14 were compared. We added MEF IF images as Fig. S1 and a correspondent text passage into the manuscript (P. 4, L. 94-97).

Were the experiments with MEFs conducted on cells derived from one single wild type and knockout mouse? After 16 passages of primary mouse fibroblasts, the cells are in fact immortalized and serial passaging effects could occur. Cells should be used at low passage numbers and/or independent biological repeats should be done.

Mouse embryonic fibroblasts of each *Abcd5*^{-/-} as well as *Abcd5*^{+/+} mice were prepared from 6-7 siblings of one litter, respectively and pooled. We added a corresponding statement to the “Methods section (P. 23, L. 553-570).

Indeed the referee is right at a passage number of 16 MEFs are in an immortalized state. However, at low passage numbers, the cell morphologies in the primary cultures are highly heterogeneous precluding a straight-forward morphological analysis of peroxisomes. However, to take into account, that the alterations in the peroxisomal DHA response in the *Abcd5*^{-/-} may be caused by further mutation during the immortalization process, the experiments were repeated 3 times with cells independently propagated from frozen *Abcd5*^{+/+} and *Abcd5*^{-/-} stocks at passages 13-15. We added a Figure showing the result from this analysis for the reviewers (Fig. R1). Please note that, *Abcd5*^{-/-} at the passage number used are still somewhat heterogeneous in size, which is shown by the difference in total peroxisomal area per cell (B) and the percentage area of peroxisomes per cell (E). To include the results from this second experimental series into the manuscript, we replaced the graphs of Fig. 1E-H with new ones summarizing the results from all experiments.

Paragraph on cerebellar phenotype : From the lipid analysis, it is concluded that there is a typical β -oxidation disorder, but the levels of DHA, that might have an effect on peroxisome size and shape, were not quantified.

ACOX1- and D-BP-deficient fibroblasts show reduced levels of DHA in membrane lipids and possess few peroxisomes (Itoyama et al. 2012). Number of peroxisomes is neither reduced in MEF, liver and cerebellum of *Abcd5*^{-/-} mice (Fig. 5, 6, Fig. S3). It should be, however, taken into account, that β -oxidation, in contrast to an ACOX1- or D-BP-deficiency, is only reduced but still functional in ACBD5-deficient cells (Ferdinandusse et al. 2017). We did not specifically determine DHA concentrations in the membrane lipids of liver and cerebellum. However, levels for Lyso-PC (22:6) and Lyso-PE (22:6), which give an estimate on the amounts of DHA in phospholipids, in liver and cerebellum (Fig. 8) do not point to reduction to an extent comparable with a complete β -oxidation deficiency.

Line 125: it is stated that phytanic acid was measured but data were not shown, nor discussed. More importantly, bile acid intermediates are particularly important substrates for peroxisomal β -oxidation in liver, and nothing is mentioned on bile acid species and levels in the manuscript. This is essential to obtain a comprehensive view on the lipid changes.

Phytanic acid measurements were included in Suppl. data 2, were they might be easily overlooked. We apologize for that. For a better visibility, representative graphs are now included as Fig. S2. Additionally, bile acid metabolites were measured in mouse urine, to analyze a potential impact of the ACBD5 knockout on peroxisomal bile acid synthesis. However, no significant differences between *Abcd5*^{-/-} and *Abcd5*^{+/+} mice were detected. Results of all measurements are included as Suppl. data 3. Additionally, a graph summarizing

the most important bile acid metabolites is shown in Fig. S2. The manuscript text was complemented accordingly (P. 6, L. 129-135).

The cerebellar phenotype was investigated in great detail but no information is given on the other brain areas, leaving the reader in doubt whether there is no pathology or whether this was not looked at. This information should definitely be included. This is also prompted by the observation that the loss of Purkinje cells seems to be mild, whereas the kyphosis seems to be pronounced, raising the question whether other pathologies occur in the nervous system. With respect to the obvious locomotor phenotype, we focused our current brain investigations primarily on the cerebellum. The strong kyphosis may point to a pronounced cerebellar defect in the spinocerebellum – a much smaller cerebellar area if compared to the pontocerebellum comprising most of the cerebellar hemispheres. Out of statistical reasons, we quantified Purkinje cell numbers in the pontocerebellum, which coordinates primarily movements of the limbs and distal parts of the body reflecting the ataxic phenotype. To better illustrate the ataxia, which gets quite prominent in older animals, we provide now two videos, showing a 1 year-old WT and *Acbd5*^{-/-} mouse during a ledge test. As we observed no other bone deformations or muscular alterations during disease progression, we conclude that the kyphosis is of neurologic origin. However alterations in further motor centers of the brain cannot be excluded and will be analyzed in the future. We added a correspondent statement to the text (P. 8, L. 196-199). To illustrate that the neurologic phenotype of the mice is not confined to the cerebellum, we added a supplemental figure showing initial data on the retinal phenotype of the mice (Fig. S4).

The title of the paragraph starting on line 198 is not well chosen. According to the conclusion it is intended to support the function of ACBD5 as an acylCoA binding protein delivering substrates to the peroxisome. The reasoning from line 241- 245 is not clear at all. It is impossible to deduce from normal ACOX1 activity that peroxisomal β -oxidation functions normally. In addition, how can it be concluded from these data that ACBD5 supplies Acyl-CoAs to the transporters? Overall, the message and the data that ACBD5 functions as a protein providing acyl-CoA to peroxisomal β -oxidation is not supported and conveyed well. It is quite unusual that the number of peroxisomes increases (proliferation) while the peroxisomal enzyme expression levels are unchanged. Are the authors aware of any other published condition with this discrepancy between peroxisome number and enzyme expression levels?

We agree that our own data presented in the manuscript does not allow concluding that ACBD5 delivers acyl-CoA for the import into peroxisomes. We observed that ACOX activities in isolated, lysed peroxisomes are comparable to the WT situation. ACOX are the rate determining enzymes in the peroxisomal β -oxidation pathway and we are not aware of post-translational modifications, which regulate β -oxidation enzyme activities at steps 2-4 of the pathway. Western blotting data on isolated peroxisomes further implies that the protein composition of isolated ACBD5-deficient peroxisomes is largely comparable.

We further drew our conclusion based on findings in human fibroblasts and HeLa cells that completely lack ACBD5 (Ferdinandusse et al. 2017, Yagita et al. 2017). Those still show normal beta-oxidation activity when supplied with high levels of substrate, but showed reduced activity when substrate levels were limiting, suggesting that VLCFA import/supply changed in response to the loss of ACBD5. However to avoid a confusion between own data and interpretation of findings from others, the corresponding text passage was removed (P. 11, L.259ff.).

With respect to the phenomenon of peroxisome proliferation observed in hepatocytes of *Acbd5*^{-/-} mice we did not conclude that peroxisomal β -oxidation activities are unchanged. We

show, that levels of peroxisomal β -oxidation enzymes are elevated in post nuclear and light mitochondrial fraction of *Acbd5*^{-/-} livers (WB Fig. 7A, B and C lanes for PNS and LM) reflecting the overall increase in peroxisomes abundance. In line, ACOX activities are elevated twice in the post nuclear supernatant (Fig. 7).

By contrast, the protein composition of individual peroxisomes is largely comparable between WT and *Acbd5*^{-/-} mice (Fig. 7). Thus, expression levels of peroxisomal proteins are increased in the latter. However, there are no extensive changes in the proteome composition at the level of single peroxisomes. Generally, peroxisomal protein changes after peroxisome proliferation are less intense at the level of isolated peroxisomes than in homogenates (the latter primarily reflect the increases in peroxisomes number). Moreover, in response to activation of PPAR α by fibrates, especially Pex11 α , L-BP and PMP70 show significant elevation at the level of isolated peroxisomes, while proteins like D-BP, ACOX1 and Pex14 do not change strongly at the organelle level (Islinger et al. 2007, Beier et al. 1988). Peroxisome proliferation in *Acbd5*^{-/-} mice is considerably less pronounced than in fibrate-treated rodents, which might explain, that L-BP and PMP70 are not significantly changed at the level of isolated peroxisomes. Proliferation of peroxisomes without specific induction of peroxisomal β -oxidation was previously reported e.g. after administration of the hypolipidemic drug BM15,766 (Baumgart et al. 1990). Likewise, the proliferation observed in *Acbd5*^{-/-} mice could be PPAR α -independent. However, we cannot exclude that a moderate PPAR α activation induced the moderate peroxisome proliferation observed, while proteome changes in isolated peroxisomes were too marginal to be detected by the applied analytical methods. As we are not able to rule out one of either theories, we modified the text of the paragraph to make that point more clear (P11, L. 256-259). To avoid the impression that we claim that the peroxisome increase is without doubt induced by activation of PPAR α the paragraph title was changed to: “The loss of ACBD5 is accompanied by elevated peroxisomes numbers in hepatocytes”

There are several changes with age in the mutant mice: body weight, lipid droplets in liver. Can this not be related to the aggravating motor disability? Was the food intake monitored? This should be incorporated in the text. Line 215: the conclusion ‘LCFA preferentially incorporated into TG for lipid storage’ is unclear.

In order to evaluate if the weight differences between *Acbd5*^{+/+} and *Acbd5*^{-/-} mice are related to the aggravating motor disability we compared food uptake and weight change in one-year old mice after access to food was alleviated (placement at cage bottom). Indeed, at this age *Acbd5*^{-/-} mice consume significantly less food than *Acbd5*^{+/+} mice when kept under standard conditions. Alleviating food accessibility did result in a trend towards increased food uptake (no significance) in both mouse strains. However, the increased food consumption did not result in a measurable weight gain in *Acbd5*^{-/-} mice after 3 weeks of alleviated food access. Nevertheless, we cannot exclude that the motor deficits generally hinder proper food uptake at this phenotype stage. A short text passage and a correspondent graph were included into the manuscript (P. 6-7, L. 146-151; Fig. S2).

Line 215: We rephrased the sentence to “The lower number of lipid droplets are further in line with the lower body weight of older *Acbd5*^{-/-} mice, pointing to a generally reduced fat storage.

The analysis of cerebellar degeneration in combination with lipidomics is very interesting. Do the authors have any idea on the absolute levels of the ULCFA in cerebellum?

Unfortunately, absolute ULCFA levels cannot be deduced from the current lipidomics measurements. The lipidomics experiment is semiquantitative and not intended for quantitative determination of absolute metabolite levels. As every compound has its own response factor in the MS and there is only one internal standard per major class it is merely possible to perform comparative lipidomics across samples, as done here.

An older publication on the level of UCLFA from the Poulos group reported a concentration below 1% of total FA in healthy rat brain (Robinson et al. 1990). Like in this study, UCLFA were preferentially found in phosphocholines. We cited this reference in the discussion section of the revised manuscript (Page 19, L. 457)

The impact of ACBD5 deficiency on ether lipid content might have to be interpreted more carefully. There was no shortage of plasmalogens in erythrocytes, and no deficiency of PC[O] and PE[O] in liver. Can the reduction of the latter species in cerebellum not have to do with reduced levels of myelin in this tissue at the age of 1 year? Myelin is indeed the most important lipid fraction in the brain. This should be reconsidered throughout the manuscript. To draw firm conclusions, one should know whether the metabolic defects precede or are a consequence of the pathology.

Plasmalogens are an important lipid subclass, especially in neuronal tissue. In the brain, like in most tissues, the main plasmalogens have an ethanolamine headgroup. The plasmalogens are part of the PE[O] group but the lipidomics measurement cannot distinguish plasmanyl- from plasmenyl species, the latter being the plasmalogens. PC[O] usually has a much lower content of plasmenyl species, most are plasmanyl species, also in the brain. The decrease of both etherphospholipid PC[O] and PE[O] species indicates a general defect rather than a specific deficiency to synthesize plasmalogens. Cerebellar etherphospholipid synthesis apparently is hampered without affecting the etherglycerolipid species. If this would be a general effect caused by lower myelin levels in the ACBD5-deficient brain one would expect to also see lower hexosylceramides (including cerebrosides and galactocerebrosides, which are abundant in myelin). However, levels are not changed, which suggest that the lower etherphospholipid levels are due to a specific defect at the level of the peroxisome and not to less myelination. In the liver, which is metabolically very different and more aimed to glycerolipids synthesis, neutral etherlipids accumulate in ACBD5 deficiency with a trend towards lower PE[O] levels (not significant though). This likely reflects the difference in metabolism where the peroxisomal ACBD5 defects apparently yields shunting of DG[O] towards TG[O] in liver (see also changes on P. 14, L 331-337).

Levels for hexylceramides are now shown in Fig S5. The aspect of an impact of the demyelination on cerebellar etherphospholipid levels is now briefly discussed (P. 14, L. 326-328, P. 20, L. 487-489).

Importantly, in my version of Fig 8, I do not find the statistical analysis that is mentioned in the figure legend. This prevents the assessment of these data for example elevation of TG[O]. It would be informative to also analyse TAG and DAG and see how they compare with DG[O] and TG[O] levels in the same tissues. Because the Y-axis of each panel of Figure 8 starts differently, attention should be drawn on this in the figure legend. Overall the reduction in LPC-22:6 is very modest, and it can hardly be claimed that its synthesis strongly depends on contact sites between ER and peroxisomes. What is the total DHA content in the tissues and in plasma?

We apologize for this neglect. Details on the statistical analysis (two-tailed, unpaired t-test) have been added to the figure legend of Fig. 8. P-values for significant alterations in lipid classes are now indicated by stars.

Minor comments

The authors should comply with the guidelines for gene and protein nomenclature in men and mice.

Gene and protein names were corrected according to the nomenclature for mice and men.

Although the methods were very extensively described, I do not find any explanation or reference how the quantification of peroxisome number, size and shape in MEFs was performed.

The methods used for quantification of MEF peroxisomes as well as other fluorescent signals were specified in the “Methods” section under “Immunofluorescence staining & microscopy” (P. 25 L. 618-620).

In addition to the MBP western blot, it would be informative to show an overview staining of the white matter in the cerebellum using MBP antibodies.

Overviews on MBP staining in cerebelli were added as a supplemental figure (Fig. S3). However, densitometric quantification of MBP intensities did not reveal a significant decrease in myelination by this method because of substantial staining variability. To substantiate our findings, we performed Western Blots for MBP in 5 additional *Acbd5*^{-/-} and *Acbd5*^{+/+} mice (Fig. R2), which corroborate the results shown in the manuscript.

Line 141: and important role

The spelling mistake was corrected.

Line 324: “only in the cerebellum...exhibited”

The sentence was changed to: “Only the cerebellum, however, exhibited significantly elevated levels of highly unsaturated ultra-long chain FA (>32:3, ULCFA)”.

Fig7A ACBD3 should be ABCD3

ACBD3 was exchanged by ABCD3 in Fig. 7A.

Presentation of the data in the results section is not always chronologically followed in the figures, in particular figures 5,6 and 7 on liver phenotype and for example lipidomics data. It would be more logic to comment on DHA/DPA prior to homeostasis of VLCFA.

In the discussion on peroxisome proliferation in liver, it is recommended to elaborate on the discrepancy between proliferation and unaltered levels of peroxisomal enzymes.

We were aware that data presentation in Figs. 5, 6 and 7 is partially not chronological in the manuscript text. In the text we decided to describe the *Acbd5*^{-/-} mice phenotype according to the main alterations observed: difference in peroxisomes numbers, alterations in the peroxisomal proteome, reduction of ER contact sites summarizing data gained from immunofluorescence microscopy, EM and immunoblotting/enzyme activity measurements. However, we wanted to avoid a mixture of data gained by different methods in the figures, which might likewise confuse future readers, e.g. in chronological figures we would have to split EM images from the resulting graphs quantifying contact sites and 3D reconstructions. We apologize that we did not find a better solution in the data arrangement.

The order of description of DHA and VLCFA lipidomics results has been arranged in the suggested order.

Reviewer #2 (Remarks to the Author):

The manuscript entitled “The neurological phenotype 1 in ACBD5-deficient mice is associated with unexpected alterations in cellular lipid homeostasis” by Darwisch and colleagues is of great importance for our understanding of organelle interaction and even more importantly to understand what happens if this interaction is disturbed. Very convincing data showing robust effects in peroxisomal alterations, biochemical defects, and cerebellar abnormality’s accompanied by a clear phenotype. The techniques used are state of the art and

the manuscript is very well written.

There are only minor comments:

1) It is a pity that there is no statistical information provided in Fig 8. It would be interesting whether or not PE-ether are equally significant reduced in liver and cerebellum – just from the graph this might well be! For the PC-ethers it is obvious without statistic. In general I would suggest to use statistical analysis to prove that the change between wt and ko is different in those lipid classes that are finally used for the argumentation of the tissue specific different lipid alterations.

Statistical information has been added to Fig. 8. See also response to the comments on Fig. 8 of reviewer 1.

2) In Fig 3A and B it should be indicated in which colour calbindin, GFAP and MBP is stained. In particular as the figure legend indicate that the arrow in B highlight axonal swellings of Purkinje cells (one in blue and one in red)?

Colors of the calbindin, GFAP and MBP signal are now indicated in the Figure legend. We further highlighted that axonal swellings were found either unmyelinated or still covered by myelin.

3) In line 213 it is referred to Fig. 8 to see the overall decreased of liver TG levels but there are only the ether TGs and these are increased.

A supplemental figure (Fig. S5A) showing graphs for TG, DG and hexylceramides in liver and cerebellum has been added.

4) In the Material section it should be indicated in which solvent and how DHA was added.

Details on DHA solubilization and application are now specified in the “Methods” section (P. 23, L. 570-573)

5) It is unclear of how many different embryonic cell lines from ko and wt have been used for the imaging. It would be appreciated if the method used for quantification in Fig. 1E to H could be described in more detail.

Acbd5^{-/-} and *Acbd5*^{+/+} MEF were prepared from seven siblings of one litter, respectively. From a P6 stock, two cell lines were cultured to immortalization (>P13) for each phenotype. We added the missing information to the “Methods” section of the manuscript (P. 23, L.553-570)

Reviewer #3 (Remarks to the Author):

The manuscript by Darwisch et al presents the characterization of a novel mutant mouse line - Acyl-CoA binding domain containing protein 5 (ACBD5) null mice - a model of ACBD5 deficiency, a rare human disorder. The pathology associated with the condition includes, but is not limited to, retina dystrophy and motor problems that are manifest in early childhood. Loss of ACBD5 expression in patients was shown to result in increased abundance of very long-chain fatty acids (VLCFAs) linked to peroxisomal β -oxidation deficits. Despite these advances the pathogenic mechanisms of ACBD5 deficiency remain unclear. Here the authors provide a comprehensive characterization of ACBD5^{-/-} mice by examining motor performance, lipidomic profile and peroxisome properties at the cellular, biochemical and ultrastructural level. Through these analyses the author show that some phenotypic abnormalities reported in patients are recapitulated in the mouse model and report additional metabolic and cellular alterations linked to the mutation.

The work is of interest in that it characterizes a new mouse strain that could be of value as disease model and beyond that for the understanding of peroxisome biology. The fact that the mutant mouse recapitulates some of the pathologies in patients gives confidence in its value as disease model. The study is fairly comprehensive, with phenotypic analyses at multiple levels. Behavioral tests to assess motor deficits are adequate to conclude that a prominent pathology linked to the condition (motor deficit, retinal dystrophy was not tested) is recapitulated in the mouse model albeit with much delayed manifestation compared to patients. Analysis of cerebellar pathology provides some evidence of a mild neurodegenerative process with Purkinje cell loss. Lipidomic studies are thorough and of value in that they identify novel tissue-specific deficits by comparing liver tissue vs. cerebellum. Furthermore analysis of peroxisome biogenesis in the liver, including at ultrastructural level, convincingly show overabundance of peroxisomes in absence of gross tissue abnormalities, an intriguing novel finding.

SPECIFIC COMMENTS

The analysis of peroxisome-ER contact sites was carried in hepatocytes but not in other cells (MEF or cerebellum). It would be of high interest to examine contact sites in the cerebellum given evident pathology and differences observed with lipidomics. These experiments would also establish whether presumptive lack of contact sites between peroxisome-ER is a general feature of all cells lacking ACBD5.

It would be doubtlessly highly interesting to analyze peroxisome-ER contacts in the cerebellum. However, peroxisomes in the brain are microperoxisomes, much less abundant than in hepatocytes and cannot be easily distinguished from other vesicular structures by EM. Moreover, as a mixed tissue, it is only meaningful to compare organelles from a particular cell type, as peroxisomes and possibly ER-peroxisome contact sites vary substantially among cell types. We, therefore, attempted to identify peroxisomes in Purkinje cells, which can be identified by their prominent morphology, by the alkaline DAB staining method used for liver in this manuscript. Unfortunately, Purkinje cell peroxisomes contain too low amounts of catalase (see Fig. S3) to be marked by the DAB staining. Moreover, the low glutaraldehyde concentrations required to preserve catalase activities, were unsuitable for a good preservation of intracellular membranes in this tissue (see Fig. R3). We were also unable to examine peroxisome-ER distances using the Tokuyasu method. Although peroxisome-like structures were labelled with an antibody directed against PEX14, poor ultrastructural preservation prevented detailed analysis (Fig. R4). It might be possible to improve the immunogold method to a quality suitable for these analyses, but unfortunately not in the time-frame of the revision.

To provide some information on the status of peroxisomes in the cerebellum of *Acbd5*^{-/-} mice, we added a supplemental figure showing immunofluorescence images from peroxisomes in correspondent cerebellar sections (Fig. S3). A quantification of peroxisomes from Purkinje cells further shows that peroxisomes abundance in these cells is not changed to a degree as observed in liver.

Nevertheless, our findings on the drastically reduced peroxisome-ER contact sites in hepatocytes confirm for the first time *in vivo* the previous observations from ACBD5 knockdown experiments in cell lines and imply that a disruption in ER-peroxisome contract sites is not an isolated effect in a specific cell type.

A particular emphasis throughout the manuscript is placed on the potential role of deficits in peroxisome-ER membrane contact sites, induced by absence of ACBD5, in cellular and metabolic perturbations. However experiments to establish causal relationship are not presented. Because of this, although an attractive hypothesis, such emphasis seems unwarranted.

We are aware that our assumptions are only a hypothesis. To make this more evident we modified the text in several passages, reduced speculative parts in the manuscript to some extent or included alternative theories (P. 2 L. 33-35, P. 4 L. 76-77, P. 16 L. 392ff – last sentence deleted, P. 16 L390-392, P. 20 L. 478-484 – text modifications and one sentence deleted, P. 20 487-488, P. 21 L. 502-505). We hope that these modifications reduced the manuscript's emphasis on the proteins tethering function. Nevertheless, in contrast to any other single enzyme deficiency, ACBD5-deficiency is caused by a protein, which in addition to its metabolic function associated with its acyl-CoA binding domain, has a unique role as a tethering protein facilitating ER contacts. According to our data, the absence of ACBD5 reduces ER contact sites *in vivo* in considerable amounts (ca. 90% in hepatocytes). Hence, we think, that mouse phenotype and detected metabolic changes should be to some extent discussed in the light of the proteins unique function as a tethering protein.

An attempt is made to examine myelination by measuring expression of PLP and MBP (Fig. 6) but these data are not appropriately quantified and therefore inconclusive.

The immunofluorescence signals for PLP and MBP were now quantified by densitometry and depicted as ratios to the actin signals. The corresponding graph has been added as Fig. 3I.

The protein composition of peroxisomes in liver was examined biochemically (Fig. 7A-C) but no statistical analysis is reported to buttress these observations (e.g. ACBD3, ACOX1, VAPB).

Unfortunately, there is no reliable peroxisomal housekeeping protein, which is consistently found in the different fractions. Thus, a densitometry-based quantification would be rather unreliable. To confirm the results we added an immunoblot series of a second independent peroxisome isolation as supplemental information (Fig. R5), which could be presented in a supplementary figure, if requested.

Expression of ACBD4 protein level, which could play a compensatory role, was not examined (only mRNA was measured).

We tested commercially available antibodies against ACBD4 (Sigma, Santa Cruz) but did not receive signals of sufficient specificity and sensitivity in either Western Blots nor IF.

Moreover, while we could specify the expression level of the tail-anchored peroxisomal isoform by qPCR (which is shown in the publication), several isoforms of ACBD4 lead to protein products of similar sizes (30-37 KDa), which would further compromise a straightforward interpretation of results obtained by IF or WBs.

REVIEWERS' COMMENTS:

Reviewer #1 (Remarks to the Author):

The authors have made substantial changes in the manuscript in response to my remarks and those of the other reviewers.

The content is now more coherent and more informative.

One remark I want to come back on is the link between hindlimb clasping, kyphosis and cerebellar dysfunction. The authors claim that clasping and kyphosis are typical signs for degenerative processes in the cerebellum. Although these can be signs of cerebellar degeneration, they can as well be a consequence of other neurological demise (eg also occurs in Alzheimer disease models). The authors now added a video on the coordination problems of a 1-year old knockout mouse, which is informative. However, this mouse seems to be much more vivid (albeit ataxic) with no obvious hunchback, compared to the picture of a 1-year-old knockout with severe kyphosis in Fig 2A, showing a lethargic mouse with front paws spread. Is there large variability in phenotype between the knockouts?

This is why I suspect that other brain areas might also be affected. In view of the many data that were already presented, and the statement that additional brain areas will be investigated in the future, I agree that no further preliminary investigation on the cerebrum is done. However, the link clasping/kyphosis – cerebellum needs to be formulated more cautiously.

Line 136-137: it is concluded that *Acbd5*^{-/-} mice present with lipid alterations typical for a peroxisomal β -oxidation disorder but bile acid levels were unaltered and C24:0 and C26:0 levels were not strongly increased. So, I would not call this typical but rather very mild.

Reviewer #2 (Remarks to the Author):

The authors have well addressed the issues raised during reviewing process, thus from my side this article is suitable for publication.

Reviewer #3 (Remarks to the Author):

Overall the authors satisfactorily addressed concerns by adding additional information and providing explanation of technical limitations when not feasible (e.g. membrane contact sites in the cerebellum).

There are some remaining issues that should be considered.

Fig. 7A-B. Statistical analysis should be provided to strengthen interpretation of results given the observed variability.

Figure S4. New data on the retina phenotype presented in figure S4 are interesting but there is no mention of these experiments/findings and interpretation in the Results section.

The non-chronological presentation of results in figures vs. text remains problematic (from figure 5 to 7). Figure 6 and 7 could be combined and re-organized: incorporating different datasets in a multi-panel figure would be helpful and less confusing.

Title. Although the manuscript reports behavioral deficits related to cerebellar dysfunction and biochemical and histological changes in the cerebellum, a significant portion of the study is dedicated to the characterization of liver/hepatocytes. Thus, while the title refers to experimental

conclusions relevant to the cerebellum it does not convey the more extensive work and findings related to the liver.

Line 129-130. A word seems to be missing.

Line 881. Typo "wwas"

Thank you very much for your positive response on our manuscript entitled "The cerebellar phenotype in ACBD5-deficient mice is associated with unexpected alterations in cellular lipid homeostasis". Again, we would also like to thank all the referees for their helpful reviews to further increase the quality of our manuscript.

Please find a revised version of our manuscript enclosed. We have incorporated the requests of the referees as indicated below.

We hope that our manuscript can now be accepted for publication in "Communications Biology".

Reviewer #1:

Remarks to the Author:

The authors have made substantial changes in the manuscript in response to my remarks and those of the other reviewers.

The content is now more coherent and more informative.

One remark I want to come back on is the link between hindlimb clasping, kyphosis and cerebellar dysfunction. The authors claim that clasping and kyphosis are typical signs for degenerative processes in the cerebellum. Although these can be signs of cerebellar degeneration, they can as well be a consequence of other neurological demise (eg also occurs in Alzheimer disease models).

The authors now added a video on the coordination problems of a 1-year old knockout mouse, which is informative. However, this mouse seems to be much more vivid (albeit ataxic) with no obvious hunchback, compared to the picture of a 1-year-old knockout with severe kyphosis in Fig 2A, showing a lethargic mouse with front paws spread. Is there large variability in phenotype between the knockouts?

This is why I suspect that other brain areas might also be affected. In view of the many data that were already presented, and the statement that additional brain areas will be investigated in the future, I agree that no further preliminary investigation on the cerebrum is done.

However, the link clasping/kyphosis – cerebellum needs to be formulated more cautiously.

We agree that the phenomena of kyphosis, locomotor deficits and hind limb clasping cannot be exclusively interpreted as a degeneration in the cerebellum. We tried to highlight this by slightly changing two text passages:

"As the locomotor deficits, kyphosis and hind limb clasping of $Acbd5^{-/-}$ mice suggested a potential cerebellar contribution to the disease pathology, ..." (P. 7, L. 152-153).

"It should be highlighted, that in addition to the degeneration in the cerebellum, pathologic processes in other brain motor centers could contribute to the locomotor phenotype ..." (P. 8, L. 194-196).

Moreover, we added statement to this paragraph, that the degenerative alterations in the retina suggest a more widespread CNS pathology (P. 8, L. 195-197).

As the referee remarked, there is a variation in phenotype severity among the mice, which is already documented by the considerable SDs data point spreading in the graph for the behavioral test (Fig. 2D). To make this clearer we added the following sentence: "As reflected by the standard deviations, disease severity can vary to some extent among age-matched individuals" (P. 6, L. 143-145).

Nevertheless, we want to point out that the kyphosis is generally a common and severe phenomenon in $Acbd5^{-/-}$ mouse, which even leads to strong malformations of the vertebrae in older as visible in CT images. However, to document the ataxic phenotype we selected a mouse with a mild kyphosis in order to avoid that the balancing disabilities of the mice could be attributed to the malformed habitus of the mouse.

Line 136-137: it is concluded that *Acbd5*^{-/-} mice present with lipid alterations typical for a peroxisomal β -oxidation disorder but bile acid levels were unaltered and C24:0 and C26:0 levels were not strongly increased. So, I would not call this typical but rather very mild. We included a statement that alterations in peroxisomal lipid metabolism are only mild: “Thus, like human ACBD5-deficient patients, *Acbd5*^{-/-} mice present at first sight with typical but rather moderate lipid alterations of a peroxisomal β -oxidation disorder” (P. 6, L. 134-135).

Reviewer #3:

Remarks to the Author:

Overall the authors satisfactorily addressed concerns by adding additional information and providing explanation of technical limitations when not feasible (e.g. membrane contact sites in the cerebellum).

There are some remaining issues that should be considered.

Fig. 7A-B. Statistical analysis should be provided to strengthen interpretation of results given the observed variability.

A T-test for the protein of interest/actin ratios was performed. Significant alterations between *Acbd5*^{+/+} and *Acbd5*^{-/-} mice are now indicated by asterisks in the revised Fig. 6D.

Figure S4. New data on the retina phenotype presented in figure S4 are interesting but there is no mention of these experiments/findings and interpretation in the Results section.

A short text passage describing retinal alterations in *Acbd5*^{-/-} mice was added to the results section: “Of note, resembling human patients⁸, a retinal degeneration characterized by reduced photoreceptor cells, increase in microglia and astrocyte activation (Suppl. Fig. 4A, B) points to more widespread pathology in the CNS of *Acbd5*^{-/-} mice.” (P. 8, L. 197-199)

The non-chronological presentation of results in figures vs. text remains problematic (from figure 5 to 7). Figure 6 and 7 could be combined and re-organized: incorporating different datasets in a multi-panel figure would be helpful and less confusing.

As suggested by the referee, we combined and reorganized Fig. 6 and 7 to a multi-panel figure (novel Fig. 6). Accordingly, figure legends and references to Fig. 6 in the manuscript were adjusted. The original Figs. 8-10 now appear as Fig. 7-9 in manuscript text.

Title. Although the manuscript reports behavioral deficits related to cerebellar dysfunction and biochemical and histological changes in the cerebellum, a significant portion of the study is dedicated to the characterization of liver/hepatocytes. Thus, while the title refers to experimental conclusions relevant to the cerebellum it does not convey the more extensive work and findings related to the liver.

We changed the title to “Cerebellar and hepatic alterations in ACBD5-deficient mice are associated with unexpected, distinct alterations in cellular lipid homeostasis” in order to reflect, that a large of the study was performed with liver tissue.

Line 129-130. A word seems to be missing.

A redundant “and” was left in the sentence. We apologize for the mistake. The sentence was corrected to “While no differences for phytanic, pristanic and behenic acid (C22:0) could be observed, *Acbd5*^{-/-} mice exhibited slightly elevated levels of lignoceric acid (C24:0) and three-fold elevations in cerotic acid (C26:0) in plasma and tissues.” (P. 6, L. 128-130)

Line 881. Typo “wwas”

The typo was corrected (now P36, L. 889).